# MORE THAN WHAT WAS CHOSEN: LLM-BASED EXPLAINABLE RECOMMENDATION BEYOND NOISY USER PREFERENCES

**Chung Park**[1*] **Hyeongjun Yun**[1,2] **Taesan Kim**[1] **Junui Hong**[1] **Dongjoon Hong**[1] **Mira Myong**[1]
**Jihoon Oh**[1] **MinCheol Cho**[1] **Kijung Park**[1] **Min sung Choi**[1] **Jihwan Seok**[1] **Jaegul Choo**[2]
[1]SK Telecom    [2]Korea Advanced Institute of Science & Technology
{cpark88kr, rpaqhdltm, ktmountain1599, secondrun3}@gmail.com,
{dongjoon.hong, mira.myong, skt.jihoon, skt.mccho}@sk.com,
{kijung.park, ms.choi, seokjh}@sk.com, jchoo@kaist.ac.kr

## ABSTRACT

Recommender systems traditionally rely on the principle of Revealed Preference (RP), which assumes that observed user behaviors faithfully reflect underlying interests. While effective at scale, this assumption is fragile in practice, as real-world choices are often noisy and inconsistent. Thus, even LLM-based recommendation models (LLM-Rec) equipped with advanced reasoning capabilities may fail to capture genuine user preferences and often produce rationales of limited persuasiveness. To address this issue, we introduce the concept of Coherent Preference (CP), which complements RP by favoring items that are logically and causally coherent with user interaction history. Building on this perspective, we propose Conflict-Aware Direct Preference Optimization (C-APO), an LLM-Rec framework that jointly optimizes RP and CP while adaptively reconciling their agreement and conflict, delivering robust recommendation performance and logically consistent rationales. We construct a unified ordering approach that combines the RP signal, based on chosen versus unobserved items, with the CP signal, which ranks items by their logical consistency with past interaction history. In this unified preference ordering, we dynamically adjust the influence of each signal depending on whether RP and CP agree or conflict, allowing the model to better capture user intent and generate more plausible recommendations. On the Amazon Review dataset, our approach consistently outperforms approximately 20 state-of-the-art baseline models in both recommendation performance and rationale quality, achieving a $1.65\times$ relative improvement in click-through rate during deployment, thereby demonstrating its practical utility. The code and dataset are available at https://github.com/cpark88/C-APO.

## 1 INTRODUCTION

Recommender systems fundamentally aim to model user preferences. Traditional approaches are based on the concept of **Revealed Preference (RP)** (Samuelson, 1938) from microeconomics, which assumes that observed user choices reliably reflect underlying preferences. Conventional recommender systems, such as collaborative filtering (CF) models, instantiate this view by learning from user–item interaction histories and have achieved remarkable performance in research contexts.

However, real-world choices are often noisy, inconsistent, or shaped by transient emotions, contextual constraints, and limited information. Consequently, the items users select do not always align with their stable or meaningful interests (Ahn & Lin, 2024). In our analysis of the Amazon Review dataset (Fig. 1), using an LLM-as-a-Judge (Gu et al., 2024) to assess logical coherence with prior behavior, roughly 30% of the chosen (ground-truth) items could not be logically explained, suggesting that RP alone may be insufficient to capture true intent.

---

*Contact: cpark88kr@gmail.com, skt.cpark@sk.com

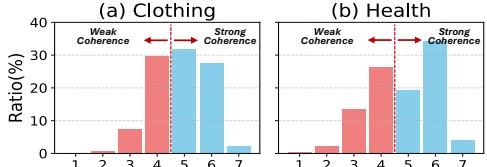
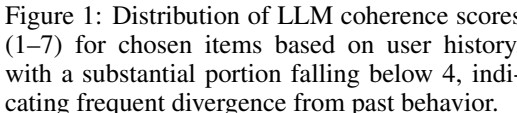
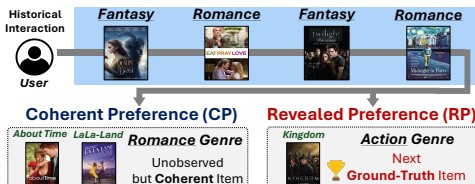

Figure 1: Distribution of LLM coherence scores (1–7) for chosen items based on user history, with a substantial portion falling below 4, indicating frequent divergence from past behavior.

Figure 2: The chosen item (*Action*) diverges from the user's romantic-history pattern, whereas the unobserved item (*Romance*) better aligns with preferences inferred from viewing history.

Consider a user who consistently watches movies in the Fantasy and Romance genres (Fig. 2). At some point, the user watches an action movie—an atypical choice given their prior viewing history. Such a choice may not reflect the true preference of users, as it may be driven by shared account usage, social viewing contexts, or transient factors such as promotional events. A romantic fantasy film—aligned in emotional tone and narrative structure with the user's history—would likely serve as a more coherent and persuasive recommendation. This motivates **Coherent Preference (CP)**: the preference over items that are causally aligned with prior behavior. Inspired by behavioral economics, CP complements RP by asking not only **what was chosen**, but **what would likely be chosen if behavior were consistent and explainable**. This perspective highlights a core limitation of the training paradigm in current LLM-based recommenders (LLM-Rec), as methods such as Direct Preference Optimization (DPO) (Rafailov et al., 2023) typically rely on the RP ordering (chosen ≻ unobserved). Consequently, despite their advanced reasoning capabilities, LLM-Rec models may not fully capture genuine user preferences and thus often produce rationales that lack persuasiveness (Tsai et al., 2024). This limitation is particularly critical in platforms such as *Instagram* or *Amazon*, where exposing both recommendations and their rationales may undermine user trust.

To address this issue, we introduce Conflict-Aware Direct Preference Optimization (C-APO), an LLM-based recommendation framework that jointly models RP and CP orderings while adaptively reconciling their agreement and conflict. While RP captures observable user-item interactions that boost recommendation performance, CP complements it by modeling the reasoning behind choices—beneficial for inferring intent and generating persuasive rationales. For each user, we construct a triplet consisting of the ground-truth chosen item and two unobserved alternatives—items the user did not interact with, which we hereafter refer to as *rejected items*. A state-of-the-art LLM is then prompted to generate, for each item, a natural-language rationale explaining its relevance to user interaction history along with a coherence score. Of the two rejected items, the one with the higher CP coherence score is labeled *hard rejected*, and the one with the lower score *easy rejected*. Each triplet element includes both the item and its post-hoc rationale, with the coherence scores further validated through human evaluation and statistical testing.

In C-APO, we first establish the RP ordering, which always ranks the chosen item above all rejected ones (i.e., *chosen ≻ hard / easy rejected*). On top of this, CP introduces a secondary ordering among the rejected items based on coherence scores (i.e., *hard ≻ easy*), yielding the unified triplet ranking *chosen ≻ hard rejected ≻ easy rejected*. However, CP is not limited to ordering among the rejected items: it can also compare the chosen item against the rejected ones. This is where the interaction between RP and CP emerges—alignment occurs when the chosen item also receives the highest coherence score, while conflict arises when a rejected item is judged more coherent, indicating that the chosen item may not fully reflect genuine user preference. Therefore, while DPO relies solely on pairwise comparisons between the chosen and rejected items based on RP, C-APO introduces a unified ordering that integrates both RP and CP over all items and adjusts their alignment and conflict. With our conflict-aware adaptive weight, it probabilistically reconciles the two signals by strengthening CP when aligned with RP, and diminishing it when conflicting. This helps the model avoid overfitting to noisy signals in RP by leveraging the reasoning-based perspective of CP.

We also provide a gradient analysis of how C-APO modulates the learning dynamics. When the chosen item shows low coherence with user interaction history—i.e., when the RP and CP orderings are in conflict—its likelihood is effectively suppressed, reducing its selection probability.

We conducted large-scale experiments across five Amazon Review domains, comparing our method against approximately 20 state-of-the-art baselines. Our method consistently outperformed most

baselines in both recommendation performance and rationale quality. Furthermore, in a real-world online A/B test, our model achieved a $1.65\times$ relative improvement in click-through rate over existing models, demonstrating its practical effectiveness. **We highlight three key contributions:**

• We introduce the Coherent Preference and present a data construction recipe for generating rationales with coherence scores, releasing the entire dataset publicly despite the high API/GPU cost.

• We develop C-APO, a reinforcement learning method that adaptively and probabilistically reconciles conflicts between RP and CP orderings, improving both generalization and rationale quality.

• Our method outperforms about 20 baselines across five Amazon Review domains and achieves significant gains in online A/B testing, demonstrating real-world applicability.

## 2  MOTIVATION AND PRELIMINARIES

### 2.1  PREFERENCE AND RECOMMENDER SYSTEM

Traditional recommenders are built upon **Revealed Preference (RP)**, which infers user interests directly from observed interactions such as past purchases or clicks. Yet, real-world choices are often noisy and shaped by transient factors, making RP an imperfect proxy for stable intent (Ahn & Lin, 2024). To complement this, we introduce **Coherent Preference (CP)**, which focuses on items that are logically consistent with prior behaviors—what users could or should prefer under coherent reasoning. CP represents a behavioral economics-inspired critique and extension of the classical RP paradigm (Hédoin, 2016), while introducing interpretability and generalizability into LLM-Rec. **While the RP signal provides a user-item interaction signal, the CP signal complements it by capturing the reasoning behind choices—an aspect that is particularly useful when modeling user intent or generating explanation rationales.**

### 2.2  PROBLEM SETUP AND TASK DEFINITION

**Definition 1: User Behavior History.** We assume a sequential recommendation task setting where a user $u \in \mathcal{U}$ interacts with a sequence of items $S_u = [i_1, i_2, \ldots, i_T]$, where each $i_t \in \mathcal{I}$ is an item from the whole item set $\mathcal{I}$. Each item $i_t$ is represented by a tuple containing the item's title and description. Then, we set the chosen item $i_c \in \mathcal{I}$ that the user interacted with next. Based on this formulation, we define the user–item interaction dataset $\mathcal{D} = \{(S_u, i_c) \mid u \in \mathcal{U}\}$. Note that we denote the ground-truth item as the *chosen* item and non-interacted items as *rejected* items.

**Definition 2: Triplet Rationale Dataset Construction Recipe.** The chosen item $i_c$ is the user's next observed choice, reflecting their revealed preference (RP). However, RP does not always exhibit logical coherence with prior behavioral patterns. Accordingly, we constructed a dataset to model coherent preference (CP). We randomly sampled two unobserved items—i.e., items the user has not interacted with—$i_1^-$ and $i_2^-$ from the whole item set such that $i_1^-, i_2^- \notin S_u \cup \{i_c\}$. For each item $i \in \{i_c, i_1^-, i_2^-\}$, we prompted a state-of-the-art LLM to generate: (1) a natural-language rationale $r$ explaining why item $i$ might be recommended given $S_u$, and (2) a coherence score $s \in \{1, 2, \ldots, 7\}$ assessing the logical consistency and persuasiveness of the corresponding rationale. This score follows a rubric ranging from 1: *Very Weak* (no connection) to 7: *Very Strong* (highly coherent and contextually perfect). In the LLM-as-a-Judge framework, this approach is referred to as single-answer grading (Gu et al., 2024). We validated the LLM-based coherence scores against human expert and annotator ratings, confirming substantial agreement with human judgments (Spearman's $\rho = 0.71$, $p < 0.0001$). All validation procedures are provided in Appendix D.

We then compared the coherence scores $s$ of $i_1^-$ and $i_2^-$ to determine the CP ordering. The item with the higher score was labeled as the *hard rejected* item $i_h$, and the one with the lower score was labeled as the *easy rejected* item $i_e$. Each constructed instance is represented as a unified ordered triplet $(i_c \succ i_h \succ i_e)$—that is, RP implies $i_c \succ i_h$ and $i_c \succ i_e$, while CP implies $i_h \succ i_e$. CP can also induce an ordering between the chosen and rejected items based on coherence scores, which is not included in the fully ordered triplet $(i_c \succ i_h \succ i_e)$ but is used to assess agreement or conflict with RP and modulate the unified ordered triplet accordingly. For each item $i$, we attached its generated rationale $r$ and the corresponding coherence score $s$, and wrote $y = (i, r, s)$; thus the instance can be expressed as the totally ordered triplet $(y_c \succ y_h \succ y_e)$, with $y_c = (i_c, r_c, s_c)$, $y_h = (i_h, r_h, s_h)$, and $y_e = (i_e, r_e, s_e)$—indicating that **our model not only learns the recommended item, as in prior work, but also the rationale supporting its recommendation.** Empirically, as shown in Fig. 3,

we found that although $i_c$ often received higher coherence scores than the rejected items, there is a nontrivial fraction of cases in which $i_c$ scores lower, highlighting a conflict between RP and CP.

Although extending beyond two rejected items is possible, it would substantially increase the cost of generating rationales and scores with the LLM as well as the training overhead (Cai et al., 2025). Since our model captures RP–CP alignment effectively with two rejected items, we leave such extensions to future work. We have released our dataset despite substantial LLM API and GPU costs [1].

**Problem Formulation.** For a given user $u$ with an interaction history sequence $S_u = [i_1, i_2, \ldots, i_T]$, the model aims to output a tuple $(\hat{i}, \hat{r})$, where $\hat{i}$ denotes the next recommended item and $\hat{r}$ is a natural-language explanation justifying the recommendation.

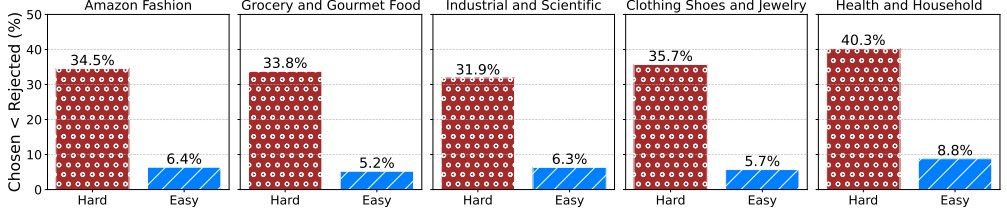

Figure 3: A non-trivial portion of cases across five domains of the Amazon Review exhibit higher coherence scores for rejected items compared to the chosen item, indicating conflicts between Revealed Preference (RP) and Coherent Preference (CP).

## 3 CONFLICT-AWARE DIRECT PREFERENCE OPTIMIZATION (C-APO)

### 3.1 RECALL OF DIRECT PREFERENCE OPTIMIZATION (DPO)

**Supervised Fine-Tuning (SFT).** In general, most alignment approaches (i.e., reinforcement learning) in recommendation, including Direct Preference Optimization (DPO) (Rafailov et al., 2023), are performed following a supervised fine-tuning (SFT) stage on recommendation-specific data. In this stage, the LLM backbone is trained with a causal language modeling loss on prompts containing user history and completions consisting of chosen items and rationales, and all subsequent descriptions are based on the tuned LLM obtained after SFT. Further details of SFT are provided in Appendix C.

**Direct Preference Optimization (DPO).** DPO does not rely on explicitly training a reward model; instead, it infers the optimal policy directly from pairwise preference data, namely *chosen* versus *rejected* items. To achieve this, the reward function $g$ is reparameterized in terms of the optimal policy through a formulation: $g(x, y) = \beta \log \frac{\pi_\theta(y|x)}{\pi_{\text{ref}}(y|x)}$, where $\pi_\theta$ is the trainable target LLM and $\pi_{\text{ref}}$ is the reference model, with the partition function omitted for simplicity. For the preference learning dataset $\mathcal{D} = \{(x_u, y_c, y^-)\}$, where $x_u$ is an input prompt constructed from user history $S_u$ (and related context), and $y_c$ and $y^-$ denote the chosen and rejected items, respectively, preference can be formulated by the Bradley–Terry model as $p(y_c \succ y^- \mid x_u) = \sigma\big(g(x_u, y_c) - g(x_u, y^-)\big)$. DPO formulates the preference likelihood as the following objective:

$$\mathcal{L}_{\text{DPO}}(\pi_\theta; \pi_{\text{ref}}) = -\mathbb{E}_{(x_u, y_c, y^-) \sim \mathcal{D}}\left[\log\sigma\left(\beta \log \frac{\pi_\theta(y_c|x_u)}{\pi_{\text{ref}}(y_c|x_u)} - \beta \log \frac{\pi_\theta(y^-|x_u)}{\pi_{\text{ref}}(y^-|x_u)}\right)\right]. \quad (1)$$

However, existing applications of DPO in recommender systems primarily rely on RP, under the assumption that the chosen item is preferred over the rejected one. To overcome this limitation, we propose **C-APO**, a novel framework that integrates both RP, which captures user–item interaction signals, and CP, which models the reasoning behind choices—thereby improving recommendation performance and enhancing the persuasiveness of generated rationales.

### 3.2 DERIVATION OF C-APO

As mentioned in Section 3.1, we are given a dataset of user preference triplets $\mathcal{D} = \{(x, y_c, y_h, y_e)\}$. To align LLM-Rec with triplet preference, we first derive the preference distribution.

---

[1] https://drive.google.com/drive/folders/1EGI8ZJ7ABjrKrniybddEA24PleR2V31M

**Plackett-Luce (PL) model** For a prompt $x_u$ and three candidate responses $\{y_c, y_h, y_e\}$, the Plackett–Luce (PL) probability of a permutation $\tau$ is as follows:

$$p(\tau \mid y_c, y_h, y_e, x_u) = \prod_{j=1}^{3} \frac{e^{\left(g_\theta(x_u, y_{\tau(j)})\right)}}{\sum_{l=j}^{3} e^{\left(g_\theta(x_u, y_{\tau(l)})\right)}}, \tag{2}$$

where $e^x$ denotes the exponential function, with the shorthand $e$ used in place of $\exp(\cdot)$ due to space constraints, and the implicit reward is given by $g_\theta(x, y) = \beta \log \frac{\pi_\theta(y|x)}{\pi_{\text{ref}}(y|x)}$. An example of the prompt used for this task is provided in Appendix E, which is also used in our C-APO framework.

For the desired full ranking $y_c \succ y_h \succ y_e$, let $\tau^\star = (y_c, y_h, y_e)$. Within $\tau^\star$, RP is formalized as $y_c \succ y_h$ and $y_c \succ y_e$, whereas CP is formalized as $y_h \succ y_e$. With shorthand $g_c = g_\theta(x_u, y_c)$, $g_h = g_\theta(x_u, y_h)$, $g_e = g_\theta(x_u, y_e)$, Eq. 2 can be rewritten as:

$$p^*(y_c \succ y_h \succ y_e \mid x_u) = p^*(\tau^\star \mid y_c, y_h, y_e, x_u) = \frac{e^{g_c}}{e^{g_c} + e^{g_h} + e^{g_e}} \cdot \frac{e^{g_h}}{e^{g_h} + e^{g_e}}. \tag{3}$$

We maximize the log–likelihood of the complete ordering; the loss function is therefore the negative log-probability:

$$\mathcal{L}_{PL} = -\mathbb{E}_{(x_u, y_c, y_h, y_e) \sim \mathcal{D}} \left[ \log p^*(y_c \succ y_h \succ y_e \mid x_u) \right]. \tag{4}$$

Substituting Eq. 3 into Eq. 4 and simplifying yields:

$$\mathcal{L}_{PL} = -\mathbb{E}_{(x_u, y_c, y_h, y_e) \sim \mathcal{D}} \left[ \underbrace{\log \sigma \left( -\log \left( e^{(g_h - g_c)} + e^{(g_e - g_c)} \right) \right)}_{\text{(1) Revealed Preference (RP)}} + \underbrace{\log \sigma \left( -\log \left( e^{(g_e - g_h)} \right) \right)}_{\text{(2) Coherent Preference (CP)}} \right]. \tag{5}$$

Eq. 5 shows that the PL model jointly enforces the full ordering in a single objective, thereby preserving permutation consistency. The first term in Eq. 5, corresponding to RP, encourages the model to rank the chosen $y_c$ higher than the rejected items $y_h$ and $y_e$. The second term, based on the coherence score, explicitly models the CP ordering between $y_h$ and $y_e$. However, the PL model does not directly model the ordering between chosen and rejected items from the perspective of CP. As a result, the first term in Eq.5, which reflects RP, cannot correct for potential noise inherent in user interactions. **Notably, the CP signal can also capture the relative ordering between chosen and rejected items via LLM-generated coherence scores. This introduces a potential source of agreement or conflict between RP and CP signals.** To this end, we propose a conflict-aware weighting mechanism that adjusts the contribution of the RP and CP signals in Eq. 5.

**Conflict-Aware Adaptive Weight** As previously discussed, while $y_c$ is always preferred over all rejected items from the RP perspective, this may not hold from the perspective of CP. In cases where the coherence score $s_c$ of the chosen item is lower than that of $s_h$ or $s_e$—indicating a conflict between RP and CP—we down-weight the relative reward of the chosen item compared to the rejected ones using a trainable weight. Conversely, when RP and CP are in agreement, the chosen item's relative reward is further emphasized. As a result, the model avoids overfitting to behavior-only signals and instead learns to generate more coherent rationales, effectively mitigating noise in the RP signal.

We define the conflict-aware reward difference as $w_{i,j}(g_i - g_j)$, where $w_{i,j}$ denotes the trainable **conflict-aware adaptive weight** assigned to each item pair $i, j \in \{c, h, e\}$. **This weight reflects the probabilistic advantage of item $i$ over item $j$ based on their coherence scores.** Then, we replace the term $(g_i - g_j)$ in Eq. 5 with $w_{i,j}(g_i - g_j)$, and rewrite this equation as follows:

$$\mathcal{L}_{\text{C-APO}} = -\mathbb{E}_{(x_u, y_c, y_h, y_e) \sim \mathcal{D}} \left[ \underbrace{\log \sigma \left( -\log \left( e^{w_{c,h}(g_h - g_c)} + e^{w_{c,e}(g_e - g_c)} \right) \right)}_{\text{(1) Conflict-Aware Revealed Preference aligned with Coherent Preference}} + \underbrace{\log \sigma \left( -\log \left( e^{w_{h,e}(g_e - g_h)} \right) \right)}_{\text{(2) Conflict-Aware Coherent Preference}} \right]. \tag{6}$$

The detailed formulation of the conflict-aware adaptive weight $w_{i,j}$ is as follows. Although these coherence scores show a statistically significant positive correlation with human ratings (Section 2.2), we treat them as noisy observations rather than gold-standard references, as even the state-of-the-art (SOTA) LLM may produce inconsistent evaluations. To calibrate them, we re-examine the same inputs used for coherence scoring—user history $S_u$, item $i$, and post-hoc rationale $r$—through a

smaller text encoder (e.g., SBERT), which provides an additional stage of validation for the scoring performed by the SOTA LLM.

Specifically, we use the LLM coherence score $s$ along with the text encoder feature $z_u = f_{\text{text}}(S_u, i, r) \in \mathbb{R}^{B \times D}$, where $f_{\text{text}}$ is a frozen text encoder, and $z_u$ denotes its output representation for a batch of size $B$ and hidden dimension $D$. Inspired by the Thurstone–Mosteller model (Thurstone, 2017; Mosteller, 1951); see Appendix J for details, we compute the mean $\mu$ and standard deviation $\tilde{\sigma}$ of the score $s$ to calibrate it as follows:

$$\mu = s + \texttt{Gate}(z_u) \cdot \texttt{FC}_1(z_u), \quad \tilde{\sigma} = \text{softplus}(\texttt{FC}_2(z_u)),$$

where $\texttt{Gate}$ is a sigmoid activation followed by a fully connected layer, and $\texttt{FC}_1$ and $\texttt{FC}_2$ are trainable fully connected layers with ReLU activation. All outputs—$\texttt{Gate}(z_u)$, $\texttt{FC}_1(z_u)$, and $\texttt{FC}_2(z_u)$—are in $\mathbb{R}^{B \times 1}$. We convert these into a soft pairwise weight:

$$w_{c,h} = \Phi\Big(\frac{\mu_{y_c} - \mu_{y_h}}{\sqrt{\tilde{\sigma}_{y_c}^2 + \tilde{\sigma}_{y_h}^2}}\Big), \quad w_{c,e} = \Phi\Big(\frac{\mu_{y_c} - \mu_{y_e}}{\sqrt{\tilde{\sigma}_{y_c}^2 + \tilde{\sigma}_{y_e}^2}}\Big), \quad w_{h,e} = \Phi\Big(\frac{\mu_{y_h} - \mu_{y_e}}{\sqrt{\tilde{\sigma}_{y_h}^2 + \tilde{\sigma}_{y_e}^2}}\Big), \tag{7}$$

where $\Phi$ is the Gaussian cumulative distribution function, mapping the weight to the $[0, 1]$ range. The resulting $w_{i,j}$ modulates C-APO updates, *amplifying* RP–CP agreements (large $w_{i,j}$) and *attenuating* uncertain conflicts (small $w_{i,j}$). By softly adjusting the loss in proportion to this preference inconsistency, the model is guided to favor recommendations where RP and CP are aligned. Note that our objective Eq. 6 strictly generalizes PL/DPO; see Appendix K.

**Gradient Analysis** We conducted gradient analysis on C-APO. Let $\Delta g_{i,j} = (g_i - g_j)$, and denote $\nabla_k = \nabla_\theta g(x, y_k)$ for brevity; then, the gradient of $\mathcal{L}_{\text{C-APO}}$ with respect to parameters $\theta$ takes the following formulation, where $s_1 = \log\left(e^{w_{c,h}\Delta g_{h,c}} + e^{w_{c,e}\Delta g_{e,c}}\right)$ and $s_2 = w_{h,e}\Delta g_{e,h}$.

$$\nabla_\theta \mathcal{L}_{\text{C-APO}} = -\mathbb{E}_{(x_u, y_c, y_h, y_e) \sim \mathcal{D}}\left[\sigma(s_1) \frac{w_{c,h}e^{w_{c,h}\Delta g_{h,c}}(\nabla_c - \nabla_h) + w_{c,e}e^{w_{c,e}\Delta g_{e,c}}(\nabla_c - \nabla_e)}{e^{w_{c,h}\Delta g_{h,c}} + e^{w_{c,e}\Delta g_{e,c}}} + \sigma(s_2) w_{h,e}(\nabla_h - \nabla_e)\right]. \tag{8}$$

The gradient of the C-APO loss in Eq. 8 is mainly governed by two components. First, the gradient for the chosen item points in the direction that increases the likelihood of $y_c$, while the gradients for the rejected items push the model to decrease the likelihoods of $y_h$ and $y_e$, respectively. The strength of this effect is scaled by $w_{c,h}$ and $w_{c,e}$. When RP and CP are aligned—indicated by large $w_{c,h}$ or $w_{c,e}$—the gradient more aggressively increases the likelihood of the chosen item, while applying weaker updates to the rejected items. Second, the $\sigma(\cdot)$ terms act as modulation factors that reflect the degree of ordering inconsistency. Specifically, $\sigma(s_1)$ increases when the reward of $y_h$ or $y_e$ exceeds that of $y_c$, i.e., when $\Delta g_{h,c} > 0$ or $\Delta g_{e,c} > 0$, and $\sigma(s_2)$ increases when the reward of $y_e$ surpasses that of $y_h$ (i.e., $\Delta g_{e,h} > 0$). This results in larger gradient magnitudes applied to push up $y_c$, encouraging the model to correct the error by increasing the likelihood of the chosen item. See Appendix I for gradient derivation details.

## 4 EXPERIMENTS

Our experiments aim to answer the following questions: **(RQ1)**: Does our model achieve superior recommendation performance over CF-Rec and LLM-Rec baselines? **(RQ2)**: Are the recommendation rationales generated by our model more persuasive than those produced by other LLM-Rec baselines? **(RQ3)**: Does incorporating coherent preferences alongside revealed preferences lead to improved recommendation performance? **(RQ4)**: Does our conflict-aware adaptive weight module lead to improvements in recommendation performance? **(RQ5)**: Is the performance of the model robust with respect to variations in the hyperparameter? **(RQ6)**: Does our model demonstrate superior performance to existing baselines in online A/B testing?

### 4.1 DATASETS

We conducted extensive experiments across five diverse domains of **Amazon Review 2023** (Hou et al., 2024), comparing our model against nearly twenty baselines to demonstrate its broad applicability (Table 1). The dataset consists of domains with practical relevance to real-world applications, including *Amazon **Fashion***, ***Grocery** and Gourmet Food*, *Industrial and **Scientific***, ***Clothing** Shoes and Jewelry*, and ***Health** and Household*.

Table 1: Statistics of Datasets.

| Dataset | #Users | #Items | Avg Length | Avg Item Purchase | Sparsity |
|---|---|---|---|---|---|
| **Fashion** | 11,028 | 59,004 | 6.54 | 1.23 | 0.9994 |
| **Grocery** | 11,334 | 6,149 | 10.76 | 21.02 | 0.9926 |
| **Scientific** | 18,275 | 7,053 | 6.69 | 17.43 | 0.9956 |
| **Clothing** | 13,766 | 107,353 | 11.83 | 1.65 | 0.9994 |
| **Health** | 21,152 | 74,784 | 8.49 | 2.50 | 0.9994 |

## 4.2 EXPERIMENTAL SETUP

**Baselines and Evaluation Settings** We assessed the performance of our model by comparing it to two categories of baselines: (1) CF-based (**CF-Rec**) and (2) LLM-based Recommendation model (**LLM-Rec**) as shown in Table 2. We adopted `Gemma-3-4B-it` (Team et al., 2025) as the LLM backbone, taking into account deployment constraints in real-world environments. However, the scaling-law trend (e.g., 1B, 4B, 12B) is observed in larger models (Appendix M). We evaluated recommendation performance using the *leave-one-out* method. For each user sequence, the last item was used for testing, the second-to-last for validation, and the rest for training. We adopted a common approach by pairing the chosen item (positive) with rejected (negative) items the user has not interacted with. These items form the candidate set and are incorporated into the prompt construction for training and inference. We focused on standard evaluation metrics, including *Hit Ratio* (**HR**) and *Normalized Discounted Cumulative Gain* (**NDCG**). All experiments were conducted using two NVIDIA H100 GPUs. For reproducibility, implementation details are described in Appendix A.

Table 2: Model performance comparison on *Amazon*, with the top two methods highlighted in **bold** and underline, respectively.

| Domain | | Fashion | | | Grocery | | | Scientific | | | Clothing | | | Health | | |
|---|---|---|---|---|---|---|---|---|---|---|---|---|---|---|---|---|---|
| Types | Models | HR@1 | HR@5 | N@5 | HR@1 | HR@5 | N@5 | HR@1 | HR@5 | N@5 | HR@1 | HR@5 | N@5 | HR@1 | HR@5 | N@5 |
| CF Rec | SASRec | 3.47 | 17.71 | 10.39 | 4.22 | 18.39 | 11.20 | 3.15 | 16.15 | 9.50 | 3.37 | 16.74 | 9.89 | 2.67 | 13.88 | 8.15 |
| | BERT4Rec | 1.62 | 6.90 | 4.19 | 4.27 | 18.97 | 11.47 | 2.46 | 13.41 | 7.80 | 2.19 | 12.14 | 7.02 | 2.45 | 12.85 | 7.53 |
| | S3Rec | 3.38 | 14.98 | 9.09 | 2.92 | 14.27 | 8.46 | 4.17 | 17.88 | 10.91 | 3.17 | 12.83 | 7.91 | 2.51 | 12.45 | 7.37 |
| | NextItNet | 1.93 | 6.31 | 4.15 | 2.89 | 15.00 | 8.97 | 4.88 | 24.64 | 14.52 | 3.35 | 13.75 | 8.45 | 2.70 | 13.36 | 7.94 |
| | SINE | 2.43 | 15.34 | 8.69 | 3.08 | 15.11 | 8.96 | 2.58 | 13.72 | 8.01 | 4.71 | 19.14 | 11.89 | 2.92 | 14.19 | 8.43 |
| | GRU4Rec | 1.63 | 8.38 | 4.85 | 3.82 | 17.13 | 10.28 | 3.13 | 15.76 | 9.30 | 3.12 | 13.33 | 8.12 | 3.88 | 17.84 | 10.84 |
| | TransRec | 2.27 | 12.14 | 6.92 | 4.79 | 18.70 | 11.62 | 5.33 | 24.43 | 14.71 | 3.36 | 13.95 | 8.56 | 2.80 | 13.61 | 8.10 |
| | LightSANs | 2.84 | 13.97 | 8.25 | 4.42 | 18.10 | 11.13 | 3.11 | 15.62 | 9.21 | 2.95 | 15.85 | 9.25 | 2.80 | 13.71 | 8.13 |
| | STAMP | 2.84 | 23.51 | 12.76 | 3.69 | 16.80 | 10.11 | 3.46 | 15.78 | 9.48 | 3.45 | 13.89 | 8.57 | 2.72 | 13.61 | 8.03 |
| | FPMC | 1.55 | 5.27 | 3.33 | 4.55 | 18.27 | 11.36 | 4.43 | 20.10 | 12.18 | 2.62 | 10.59 | 6.55 | 2.95 | 13.39 | 8.06 |
| LLM Rec | Rec-SAVER | 4.27 | 24.41 | 14.34 | 5.73 | 18.04 | 12.19 | 6.10 | 17.41 | 12.05 | 5.66 | 22.03 | 13.89 | 2.86 | 14.65 | 8.61 |
| | SumRec$_{DPO}$ | 8.06 | 28.65 | 18.55 | 3.84 | 15.19 | 9.89 | 4.62 | 14.43 | 9.90 | 3.97 | 18.74 | 11.82 | 3.27 | 15.10 | 9.31 |
| | RecLM | 5.86 | 9.77 | 6.62 | 5.52 | 10.60 | 8.56 | 2.59 | 7.13 | 5.96 | 2.25 | 7.08 | 5.51 | 1.90 | 6.01 | 5.13 |
| | S-DPO | 5.22 | 23.78 | 14.38 | 6.57 | 18.47 | 12.91 | 7.57 | 20.60 | 14.84 | 4.16 | 19.37 | 11.75 | 2.76 | 15.71 | 9.28 |
| | GRAM | 5.88 | 17.12 | 11.54 | 1.75 | 5.22 | 3.52 | 4.26 | 11.85 | 8.11 | 3.34 | 11.76 | 7.54 | 4.55 | 13.39 | 9.00 |
| | Intuitor | 4.46 | 10.46 | 7.71 | 6.34 | 11.58 | 8.92 | 7.10 | 16.39 | 12.57 | 3.22 | 9.53 | 6.43 | 3.30 | 8.62 | 6.19 |
| | Rec-R1 | 8.43 | 29.29 | 18.80 | 5.65 | 17.95 | 11.94 | 4.82 | 20.67 | 13.44 | 5.62 | 21.88 | 14.06 | 2.33 | 14.24 | 8.03 |
| | SLMRec | 3.21 | 6.58 | 4.87 | 3.44 | 11.11 | 7.34 | 3.13 | 24.66 | 13.75 | 3.19 | 17.71 | 10.12 | 2.77 | 12.67 | 7.63 |
| | LLMEmb | 3.22 | 7.71 | 5.15 | 3.59 | 14.61 | 8.18 | 4.11 | 13.88 | 8.57 | 3.41 | 16.80 | 9.97 | 2.48 | 7.59 | 6.39 |
| | **Ours** | **9.47** | 28.22 | **18.87** | **6.90** | **19.47** | **13.62** | **12.22** | 22.74 | **17.97** | **7.11** | **23.12** | **15.09** | **4.83** | **18.21** | **11.73** |

## 4.3 PERFORMANCE EVALUATION (RQ1)

In Table 2, we reported the performance of all the baselines and our model. **Our model consistently outperformed state-of-the-art CF-Rec and LLM-Rec models across all domains, demonstrating superior overall recommendation performance**. Specifically, our model achieved relative HR@1 gains over the second-best model in each domain, with gains of **+12.34%** (*Fashion*), **+5.02%** (*Grocery*), **+61.43%** (*Scientific*), **+25.62%** (*Clothing*), and **+24.48%** (*Health*). In addition to HR@1, our model consistently demonstrated superior performance across most domains on HR@5 and NDCG@5. This suggests that the enhanced generalization capability of the model is attributable to the integration of CP alongside RP. Even when replacing the LLM backbone with `Qwen-3-4b-Instruct` (Team, 2025) for both our model and S-DPO—the second-best baseline—C-APO achieved **15.38%** and **12.29%** higher HR@5 and NDCG@5, respectively, demonstrating that the performance improvement originates from our training approach rather than the choice of the LLM backbone.

## 4.4 EVALUATING RATIONALE QUALITY (RQ2)

We conducted a quantitative evaluation of the rationales produced by our model, applying an identical prompt across five baselines: `Gemma-3-4B-it`, `Gemma-3-12B-it`, Rec-SAVER, DPO, and S-DPO. From each of the five domains, 300 samples were randomly chosen, yielding 1,500 evaluation cases in total. Our assessment followed a two-stage

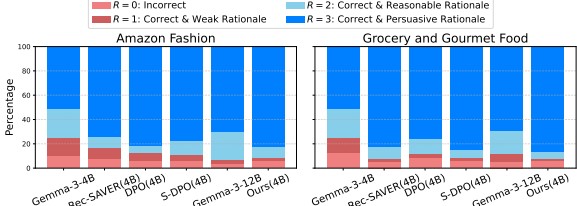

Figure 4: LLM-as-a-Judge of Rationale Quality

protocol: we first checked whether the recommended item belonged to the valid item vocabulary, and subsequently examined the plausibility of the accompanying rationale. Following the evaluation protocol of Wang et al. (2022), ChatGPT was used as an evaluator with a four-point scale: **0** = Incorrect (Hallucination), **1** = Correct but Weak Rationale, **2** = Correct with Reasonable Rationale, and **3** = Correct with Persuasive Rationale. Importantly, the evaluator LLM (e.g., `gpt-4o-series`) differs from the LLM used to generate rationales in the training data, preventing potential evaluation bias. To assess reliability, we compared LLM-based judgments with human annotations on 250 sampled examples and found statistically significant agreement (Quadratic Weighted Kappa QWK = 0.75, $p < 0.0001$). As illustrated in Fig.4, our approach consistently generated rationales that not only aligned with the correct recommendation but also enhanced its persuasiveness. Our model achieved the highest share of top-rated rationales (score 3), reaching 84.33%—representing an improvement of approximately 5.99%p over the second-best method, Rec-SAVER (78.34% → 84.33%; Fig.4). Details of the LLM-as-a-Judge prompt for rationale evaluation, the human evaluation protocol with statistical correlation analysis, and representative rationale examples are all provided in Appendix F.

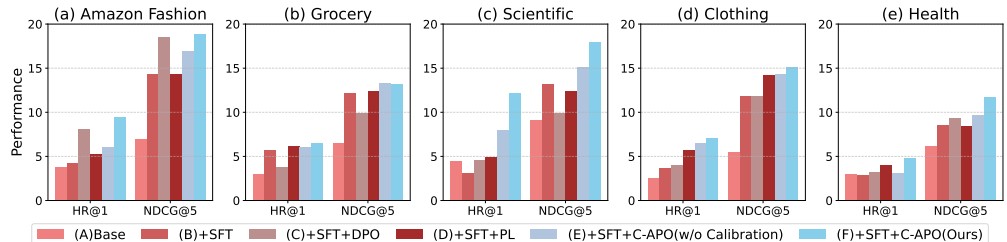

Figure 5: Comparisons between model variants across five domains

## 4.5 ABLATION STUDY

**Component Analysis** In Fig. 5, we evaluated the effectiveness of our components through five primary model variants: **(A) Base**: the `Gemma-3-4B-it` model; **(B) +SFT**: the base model with supervised fine-tuning on the recommendation dataset; **(C) +SFT +DPO**: the SFT model further aligned with DPO; **(D) +SFT +PL (DPO$_{PL}$)**: the SFT model further optimized with a Plackett–Luce objective enforcing $y_c \succ y_h \succ y_e$; **(E) +SFT +C-APO** $w/o$ **SBERT calibration**: C-APO without SBERT-based conflict-aware weighting; **(F) +SFT +C-APO**: our proposed model.

**(1) Recommendation-Specific Training Needed:** The base model (A) underperformed most variants across domains, underscoring the need for recommendation-specific training.

**(2) Preference-Based Alignment Matters:** The model (B), which is supervised fine-tuned on the recommendation-specific dataset using only chosen items, exhibited lower performance in most domains than models further aligned with preference-based objectives (C–F). This result demonstrates that preference-based reinforcement learning provides additional benefits in recommendation tasks.

**(3) Coherent Preference Improves Performance (RQ3):** Comparing the variant (C) with (F), we observe that jointly modeling CP yields better performance than relying solely on RP. This result empirically indicates that complementing RP with CP promotes improved generalization in recommendation models, demonstrating that RP alone cannot fully capture human preferences and highlighting the necessity of CP.

**(4) Conflict-Aware Adaptive Weight Improves Performance (RQ4):** The variant model (D) corresponds to the objective in Eq. 5, namely the Plackett–Luce (PL) model, which captures the full ordering of chosen and rejected items from both RP and CP perspectives. By comparing this variant with our model (F), we isolated the effect of joint RP–CP modeling versus additionally calibrating their conflicts and agreements—that is, the contribution of our Conflict-Aware Adaptive Weight module. As shown in Fig. 5, model (D) generally underperformed our approach, underscoring the effectiveness of our adaptive calibration in handling RP–CP alignment and conflict.

To further separate the effect of the SBERT-based calibration module from the conflict-aware adaptive weight itself, we also consider the variant (E) that removes the calibration step and relies solely on raw LLM coherence scores (pairwise difference followed by sigmoid) when forming the weights $w_{i,j}$. This variant assesses how well LLM-derived soft weights perform without any text-encoder-

based adjustment. As shown in Fig. 5, this model performs better than the variant (D) but worse than the variant (F), indicating that: (i) LLM coherence signals provide meaningful preference cues on their own, while (ii) the SBERT-based calibration and conflict-aware adaptive weighting jointly offer the major performance gains, especially under noisy or misaligned CP signals.

**Study on Value of $\beta$** The deviation of the LLM from the base reference policy is controlled by a hyperparameter $\beta$ in C-APO. We selected the value of $\beta$ from $\{0.1, 0.5, 1, 2, 5\}$ to explore the effect of $\beta$ on C-APO. As indicated in Fig. 6, while model performance was generally robust with respect to $\beta$, increasing $\beta$ improved performance up to 1, beyond which performance degraded.

**Scaling Law** We observed a consistent scaling-law trend (e.g., 1B, 4B, 12B). All three evaluation metrics—HR@1, HR@5, and NDCG@5—improved monotonically as the number of model parameters increased, reflecting a clear scaling effect in recommendation performance (Appendix M).

**Study on Reward of Chosen, Hard Rejected, and Easy Rejected** We visualize the relationship between the coherent score gap $\Delta s_i = s_c - s_i$, $i \in \{h, e\}$ and the conflict-aware adaptive weight $w$ using binned distributions (Fig. 7). Negative gaps ($\Delta s \leq 0$) indicate *conflicting* pairs, while large positive gaps ($\Delta s \gg 0$) indicate *aligned* pairs. As $\Delta s$ increases, $w$ increases monotonically, indicating that C-APO strengthens the relative reward of the chosen item over the rejected item when CP aligns with RP, and weakens it when they conflict. Importantly, some variability remains within each $\Delta s$ bin, since $w$ is not a direct mapping of LLM coherence scores $s$ but is calibrated through an additional text encoder. This indicates that subtle discrepancies exist between the raw coherence scores produced by the LLM and the adjusted values derived through the auxiliary text encoder.

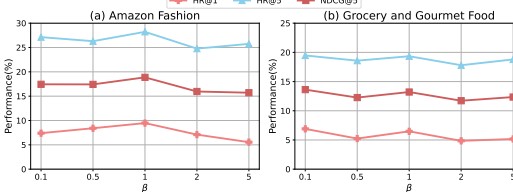

Figure 6: Study of $\beta$ on Amazon Dataset

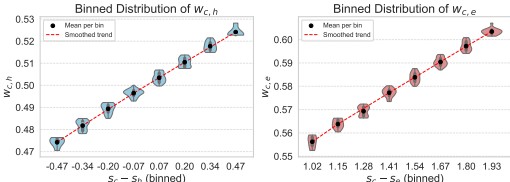

Figure 7: Relationship between Coherence Score Gap ($\Delta s = s_c - s_i$) and Calibrated Weight $w$

## 4.6 ONLINE A/B TEST (RQ6)

We deployed our model in a real-world production environment and conducted an online A/B test as a pilot study over August and November 2025. In this setting, the top-1 recommendation was presented to each

Table 3: Online A/B Test Results

| Model | # Impression | # Clicks | CTR | CVR | Latency |
|---|---|---|---|---|---|
| **Random** | 180.08K | 17.90K | 9.90% | 1.58% | 94ms/call |
| **ML model** | 127.83K | 11.94K | 9.34% | 2.01% | 120ms/call |
| **SFT** | 6.50K | 0.88K | 13.56% | 1.81% | 132ms/call |
| **Ours** | 79.50K | 11.95K | 15.03% | 2.60% | 138ms/call |

customer, accompanied by a rationale explaining the recommendation. We employed a customized version of our model tailored to business requirements. For comparison, we used three baselines: a random control, a traditional machine learning (ML) model, and an SFT-based model that also generates both recommendations and rationales. Users were randomized via a stable user-level hash into four buckets with a 4:3:1:2 traffic split (Random, ML, SFT, Ours). Note that the SFT-based model had been deployed in the online test for only about two weeks at the time of evaluation; thus, although its bucket allocation was 10%, its total impressions appear lower than other buckets. As shown in Table 3, our model achieved a +60.88% CTR lift over the ML baseline. A one-sided two-proportion $z$-test confirms that this improvement is highly significant ($Z = 39.42$, $p < 0.001$), with a 95% confidence interval (CI) of [5.40%, 5.97%] for the absolute CTR difference. This demonstrates a robust performance gain while maintaining low inference latency, confirming the model's suitability for real-world deployment.

To control for the possibility that performance gains might arise merely from the novelty of displaying rationales, we introduced an SFT-based model that also generates both recommendations and rationales but does not use conflict-aware weighting. Since the SFT model likewise presents rationales to users, it serves as a control for rationale exposure, enabling a clean isolation of improvements attributable to our method. Even under this stricter comparison, our method achieved a statistically significant CTR improvement of +1.47%p over the SFT model (15.03% vs. 13.56%; $Z = 3.20$, $p < 0.001$, 95% CI = [0.57%p, 2.37%p]). We also evaluated the conversion rate (CVR)—the

ratio of purchases to impressions—and our model achieved a statistically significant improvement over all baselines.

## 4.7 Time Complexity

**Theoretical Analysis** Let $C_{\text{LLM}}$ be the computational complexity of the backbone LLM, $B$ the batch size, and $M = 1 + R$ the number of responses (one chosen, $R$ rejected; $R = 2$). During training, C-APO computes log-probabilities for all $M$ responses by performing pairwise comparisons under a full ordering, leading to a complexity of $\mathcal{O}\big(B \cdot M \cdot C_{\text{LLM}}\big)$. Note that the training complexity of DPO is $\mathcal{O}(2B \cdot C_{\text{LLM}})$. The calibrator relies on a lightweight text encoder (109M parameters), and its computation is negligible compared to the LLM backbone, thus omitted from complexity analysis.

**Empirical Analysis** Compared to GRPO-based Rec-R1, our model trains **3.14× faster**, though it is **1.11×** and **1.26× slower** than DPO and S-DPO, respectively. This slight slowdown is expected since DPO essentially leverages only pairwise data, whereas our method adopts a triplet structure; however, the gap is negligible and justified by the performance improvements. When comparing the PL model to ours, the difference is minimal (1.001×), indicating that the additional calibration step contributes almost no overhead relative to the intrinsic time complexity of the LLM (Appendix L).

## 5 Related Work

**Collaborative Filtering-based Recommendation**

Collaborative filtering (CF)-based approaches have long formed the foundation of recommendation systems, achieving reliable performance across various practical domains. SASRec (Kang & McAuley, 2018) employs self-attention mechanisms to model sequential user-item interactions. Recurrent models such as GRU4Rec (Hidasi et al., 2015) and STAMP (Liu et al., 2018) are designed to capture both short- and long-term behavioral patterns. In addition, convolutional models like NextItNet (Yuan et al., 2019) and modality transfer frameworks such as TransRec (He et al., 2017) offer alternative approaches for encoding user–item dynamics. Despite their effectiveness in capturing historical interactions, these methods provide limited explanatory capability.

**Large Language Model-based Recommendation**

Recent work has investigated the application of large language models (LLMs) in recommendation tasks to enhance contextual understanding and explanation generation. For instance, Rec-SAVER (Tsai et al., 2024) conditions on user context to elicit rationales for recommendations and jointly optimizes the explanation and ranking objectives. RDRec (Wang et al., 2024) introduces a two-step procedure: generating rationales from user history and reviews, followed by training on sequential recommendation. Rec-R1 (Lin et al., 2025) applies Group Relative Policy Optimization (GRPO), assigning feedback based on alignment between predicted and target items. S-DPO (Chen et al., 2024) builds on DPO with a softmax-based formulation, supporting multiple negative samples and improving preference ranking. See Appendix B for LLM-Rec baseline details.

## 6 Conclusion and Future Work

We presented C-APO, a principled framework that jointly models revealed and coherent preferences. This approach effectively reconciles agreements and conflicts between the two preference dimensions, leading to improved recommendation performance and more persuasive rationales. Furthermore, extensive experiments and real-world deployment demonstrate its practical effectiveness and real-world applicability. Although C-APO shows strong performance in both offline benchmarks and real-world deployment, several limitations suggest promising avenues for future work. First, triplet-level dataset construction relies on costly LLM-based coherence scoring, which may limit retraining frequency. To address this, we are developing a lightweight distilled evaluator that approximates the full scorer at significantly reduced cost. Second, our experiments focus on text-only domains. Extending C-APO to multimodal settings—such as video, music, or image-based platforms—by incorporating vision or audio encoders is a natural next step.

ACKNOWLEDGMENTS

We would like to thank SK Telecom for providing the GPU cluster and associated research infrastructure that enabled this work, as well as for their support in integrating and validating our approach in a production service.

This work was supported by Institute for Information & communications Technology Planning & Evaluation (IITP) grant funded by the Korea government(MSIT) (RS-2019-II190075, Artificial Intelligence Graduate School Program (KAIST)). This work was also supported by the National Research Foundation of Korea(NRF) grant funded by the Korea government(MSIT) (No. RS-2025-00555621).

**Ethics statement** This work introduces a novel training objective for LLM-based recommenders by integrating both revealed and coherent preferences in a unified generative framework. Our method is designed to enhance user-aligned recommendations and transparent rationales without engaging with sensitive attributes or user-identifiable data. We foresee no potential risks of societal harm or ethical concerns, and confirm full compliance with the ICLR Code of Ethics.

**Reproducibility Statement** All experimental results are fully reproducible. To facilitate verification and further research, we provide the complete implementation code (`https://github.com/cpark88/C-APO`) and a curated dataset constructed using the ChatGPT API and `Gemma-3-27B-it`. The dataset comprises structured user purchase histories, candidate item sets, labeled positives and negatives, LLM-generated rationales, and coherence scores across multiple domains. It was produced through resource-intensive inference pipelines and rigorous human curation. Detailed hyperparameter settings are reported in Appendix A.

**GenAI Usage Disclosure** The authors used Generative AI (GenAI) tools solely for the purpose of correcting minor grammatical errors in the manuscript. No part of the research process—including problem formulation, data analysis, model design, coding, experimentation, or interpretation of results—was conducted using GenAI tools. In addition, no GenAI assistance was used in the actual writing or generation of content for this paper, aside from grammar refinement.

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

# Appendix

## A   REPRODUCIBILITY AND IMPLEMENTATION DETAILS

**Code and Dataset**   All experiments were run on two NVIDIA H100 GPUs. To support reproducibility, the implementation code is available at the anonymous links provided. We also release a dataset constructed using the ChatGPT API and `Gemma-3-27B-it`, which includes well-structured user purchase histories, candidate item lists, chosen and rejected items, LLM-generated rationales, and coherence scores across all domains. This dataset was developed through resource-intensive and costly LLM inference pipelines, coupled with extensive human curation, and is publicly released to facilitate further progress in explainable recommendation research. It comprises three subsets: a training set for supervised fine-tuning (SFT), a training set for Conflict-Aware Preference Optimization (C-APO), and a test set for evaluating model performance. Although the SFT and C-APO training sets share the same users, they differ in the rationales used, as each was generated using a different LLM, thereby yielding two distinct datasets tailored for their respective training objectives.

■ **Code**: https://github.com/cpark88/C-APO

■ **Dataset (Five Domains of Amazon Review Dataset)**: https://drive.google.com/drive/folders/1EGI8ZJ7ABjrKrniybddEA24PleR2V31M?usp=share_link

**Hyperparameters**   Table 4 lists the hyperparameters used in all experiments, including learning rate, batch size, optimizer type, warm-up ratio, gradient clipping, weight decay, and the number of epochs. We used `Gemma-3-4B-it` (Team et al., 2025) as the LLM backbone. We first performed supervised fine-tuning (SFT) using only the chosen items. During the SFT stage, we used the same hyperparameter settings as C-APO (Table 4), except for setting the number of epochs to 2 and the batch size to 4. In contrast, C-APO is trained using not only the chosen item but also the two rejected items jointly. C-APO was trained for 3 epochs with a batch size of 6, using the 8-bit paged AdamW optimizer and a cosine learning rate schedule (initial learning rate $5.0 \times 10^{-5}$, decayed to a minimum value). We adopted gradient accumulation with a factor of 8, a warm-up ratio of 0.05, and a maximum sequence length of 12k tokens. Training was performed with DeepSpeed ZeRO-2 using bf16 mixed precision, gradient checkpointing. Parameter-efficient fine-tuning (PEFT) was conducted via LoRA (Hu et al., 2022) with $r = 8$, $\alpha = 16$, and dropout $= 0$, applied to the $v$-, $q$-, $o$-, $gate$-, $up$-, and $down$- projection modules. We searched over $\beta \in \{0.1, 0.5, 1, 2, 5\}$ and report the best test performance. We ultimately set $\beta = 1$ for most domains based on validation performance. To calibrate the coherence scores generated by the LLM, we employed SBERT (Reimers & Gurevych, 2019) as the text encoder (109M parameters; `sentence-transformers/all-mpnet-base-v2`).

Table 4: Training hyperparameters for our model

| Hyperparameter | Value |
|---|---|
| Batch size | 6 |
| Epochs | 3 |
| Optimizer | Paged AdamW (8-bit) |
| Learning rate | $5.0 \times 10^{-5}$ (cosine schedule, min LR applied) |
| Warm-up ratio | 0.05 |
| Gradient accumulation steps | 8 |
| Max sequence length | 12,000 |
| Weight decay | 0.01 (default) |
| Gradient clipping | 1.0 (default) |
| LoRA rank ($r$) | 8 |
| LoRA $\alpha$ | 16 |
| LoRA dropout | 0 |
| LoRA target modules | $v$-, $q$-, $o$-, gate-, up-, down-projections |
| Precision | bf16 |
| Gradient checkpointing | True |
| Parallelism | DeepSpeed ZeRO-2 |
| Hardware | $2 \times$ NVIDIA H100 80GB GPUs |

**Evaluation Protocol**   We follow a standard leave-one-out protocol: for each user sequence, the last item is used for *test*, the second-to-last for *validation*, and the rest for *train*. At evaluation time, the *candidate set* consists of the ground-truth chosen item paired with unseen 29 rejected items, and this candidate list is *inserted into the prompt* for inference. For performance evaluation (e.g., HR@k), we set the decoding temperature to 0.7 and, when $k = 5$, generate 5 samples per prompt.

## B   DETAILS OF LLM-BASED RECOMMENDATION BASELINES

For collaborative filtering-based recommendation models (CF-Rec), we implemented all methods using the RecBole framework (Zhao et al., 2022)[2]. The LLM-based recommendation baselines are described in detail below.

---

[2] https://github.com/RUCAIBox/RecBole

■ **Rec-SAVER** (Tsai et al., 2024) performs instruction tuning using paired recommendation items and rationales, showing that this approach improves recommendation quality in both zero-shot and fine-tuning settings. We implemented this model from scratch.

■ **S-DPO** (Chen et al., 2024) introduces a ranking-oriented training objective for LLM-based recommendation, extending DPO with a softmax formulation that accommodates multiple negative samples. This approach improves the model's ability to differentiate user-preferred items and strengthens the impact of hard negative signals during optimization. This model was developed entirely from scratch by our team.

■ **Intuitor** (Zhao et al., 2025) introduces a reinforcement learning framework that eliminates the need for external rewards by leveraging the model's own prediction confidence—referred to as self-certainty—as the training signal. Based on the Group Relative Policy Optimization (GRPO) paradigm, Intuitor enables large language models to optimize their behavior using intrinsic feedback, without relying on labeled supervision. We implemented the model with reference to the official code repository [3].

■ **Rec-LM** (Lu et al., 2024) improves LLMs' instruction-following ability in recommendation by combining supervised training on recommender-labeled tasks with a reinforcement learning alignment procedure (Proximal Policy Optimization; PPO), enhancing response controllability without sacrificing recommendation performance. Our implementation was guided by the official repository provided by the authors [4].

■ **Rec-R1** (Lin et al., 2025) applies the GRPO framework to the recommendation task, where the reward is defined based on whether the generated item matches the ground-truth. In our implementation, we extend this framework by incorporating additional reward signals from collaborative filtering models such as SASRec. We implemented this model based on the repository that provides a GRPO implementation [5].

■ **GRAM** (Lee et al., 2025) formulates recommendation as a text-to-text generation task and introduces two key modules to overcome limitations in capturing item relationships and handling verbose item descriptions. Specifically, it translates semantic and collaborative signals into lexical tokens via a semantic-to-lexical translation module, and fuses multi-granular item prompts at the decoding stage through a multi-granular late fusion mechanism. We implemented the model with reference to the official code repository [6].

■ **SLMRec** (Xu et al., 2024) investigates the redundancy of deep layers in LLMs for sequential recommendation and proposes a distilled small language model optimized for efficiency. Through extensive experiments, the authors find that many intermediate LLM layers are unnecessary for performance, and design a compact model that retains effectiveness via simple knowledge distillation. We referred to the official implementation when reproducing the model [7]. Due to constraints in the official implementation, we employed a LLaMA-series backbone. Specifically, we used `meta-llama/Llama-3.2-3B`, which is comparable in scale to `Gemma-3-4B-it`, the backbone adopted in our main experiments.

■ **LLMEmb** (Liu et al., 2025) leverages LLMs to generate item embeddings for sequential recommendation, addressing the long-tail problem. It combines Supervised Contrastive Fine-Tuning (SCFT) with Recommendation Adaptation Training (RAT) to align embeddings with collaborative signals. We based our implementation on the official codebase released by the authors [8].

■ **SumRec$_{\text{DPO}}$**(Tajiri & Inaba, 2025) is trained via Direct Preference Optimization (DPO) using LLM-generated dialogue summaries and item recommendation information. While the original setting targets multi-turn conversational recommendation, we adapted the model to our single-turn sequential recommendation task by applying both SFT and DPO accordingly. Our implementation was done from scratch.

---

[3] https://github.com/sunblaze-ucb/Intuitor
[4] https://github.com/microsoft/RecAI/tree/main/RecLM-gen
[5] https://github.com/huggingface/open-r1
[6] https://github.com/skleee/GRAM
[7] https://github.com/WujiangXu/SLMRec
[8] https://github.com/liuqidong07/LLMEmb

## C  SUPERVISED FINE-TUNING PRIOR TO C-APO

**Training**   Modern LLM-based recommendation systems frequently employ supervised fine-tuning (SFT) to improve their task-specific capabilities. This process typically consists of two phases: (1) transforming recommendation datasets into textual prompt–response formats, and (2) adapting the LLM using these structured pairs. In the data preparation step, a prompt $x_u$ is created for each user $u$, which includes their interaction history $S_u$, a candidate item set $C$, and an instruction specifying the sequential recommendation task. This prompt is paired with an output $y_c$, which contains the title of the selected item $i_c \in C$ and its corresponding rationale $r_c$, forming the training example $(x_u, y_c)$. Roughly 70% of these SFT training instances are reused in the later C-APO stage, where additional information—such as rejected items, their rationales, and coherence scores—is incorporated to model user preferences more effectively. In the second phase, the LLM-based recommendation model $f_\theta$ is fine-tuned using the constructed pairs $(x_u, y_c)$, with training guided by a causal language modeling loss. This objective, widely used in language modeling, encourages the model to generate the expected output by predicting each token sequentially given the previous context, effectively casting recommendation as a next-token prediction problem. Formally, the objective of optimizing the LLM-based recommender $f_\theta$ with pair data $(x_u, y_c)$ can be formulated as:

$$\max_\theta \sum_{(x_u, y_c)} \sum_{t=1}^{|y_c|} \log(P_\theta(y_c)_t | x_u, (y_c)_{<t}), \tag{9}$$

where $|y_c|$ is the number of tokens in $y_c$, $(y_c)_t$ is the $t$-th token of $y_c$ and $(y_c)_{<t}$ is the tokens preceding $(y_c)_t$.

However, recommendation tasks are essentially user preference alignment tasks, as formalized in the above task formulation, and differ from language modeling tasks that consider only chosen item. Such a gap necessitates further exploration into aligning LLM-based recommenders with user preference, an area that has been underexplored.

**Prompt Example**   The prompt used for SFT is designed to encourage the model to select the most appropriate item from a list of recommendation candidates, given a user's purchase history, and to generate a rationale explaining the recommendation.

---

**#Prompt $x_u$**
Based on **[Purchase History]**, use your logical reasoning process to identify the most suitable item for this customer from the **[Candidate List]**. Then, include your reasoning inside the `<think></think>` tag and the recommended item inside the `<item></item>` tag.
**[Purchase History]** (1) Title: Finejo women office ladies 3/4 sleeve neck business bodycon dress, yellow, xx-large (2) Title: Farktop women's V neck long sleeves digital graffiti printed prom party maxi long dress with belt,...(truncated) $\leftarrow S_u$
**[Candidate List]**: (omitted for brevity)

**#Response $y_c$**
`<think>`The customer has previously purchased two dresses - a business bodycon dress and a long maxi dress. ... (truncated) `</think>`$\leftarrow r_c$
`<item>`Fibo steel 10 pcs women black velvet choker necklace for girls lace choker tatto necklace `</item>`$\leftarrow i_c$

---

## D  TRIPLET RATIONALE DATASET CONSTRUCTION RECIPE

As described in Section 2.2, we construct training data in the form of $y = (i, r, s)$, where each sample $y$ consists of a chosen item, two rejected items, and their corresponding rationales $r$ and coherence scores $s$ with respect to the user's interaction history. These are generated using a state-of-the-art (SOTA) LLM, based on the relevance and logical consistency between each item and the user's history. Note that the chosen item is not guaranteed to exhibit higher coherence than the rejected items, as shown in Fig. 9. We detail the data construction recipe and the subsequent human cross-checking procedure to validate its quality.

**Step1:  Rationales Generation and Evaluation**   We first prompted the state-of-the-art LLM (`Gemma-3-27b-it`) to generate post-hoc rationales separately for each of the three items—the chosen item and two rejected (i.e., non-chosen) items—based on their semantic relevance to the user's interaction history. For this tasks, we used a deterministic setup with temperature=0, following common practice in LLM-as-a-judge literature. We intentionally adopt this conservative setting because coherence scores act as supervisory signals; introducing stochasticity would inject noise directly into the optimization objective and undermine training stability. The following is an example of the prompt used to generate rationales and coherence score.

> **#Prompt**
>
> (System Prompt) You are a recommendation expert tasked with generating a rationale for the next item purchased by a customer, based on their prior behavior. Your explanation should be grounded in the causal relationship between the customer's previous actions and the actual next item. The customer will purchase the **[Next Item]** from the list of candidate items based on their given purchase history. Based on this, please follow the instruction below.
>
> **[Purchase History]** (1) Title: Bai coconut flavored water, cocofusions variety pack (2) Title: Organic coconut oil,...(truncated) $\leftarrow S_u$
> **[Candidate List]**: (omitted for brevity)
> **[Next Item]** Title: Wild planet albacore wild tuna with sea salt, canned tuna, non-gmo, kosher $\leftarrow i_c$ or $i_h$ or $i_e$
>
> (Instruction) Step by step, explain why the customer will purchase the [Next Item] based on its [Purchase History]. Describe it in a way that, as a result of the reasoning process, the next item is recommended. In particular, when deriving the reasoning process, carefully examine the [Purchase History], which includes a rating-one of Terrible, Bad, Okay, Good, or Excellent-for purchased item, along with an associated review. Analyze both the positive and negative reviews, thoroughly.
> Think about the logical relevance (reasoning process) between the customer's [Purchase History] and the [Next Item]. Also, evaluate the logical coherence between the customer's [Purchase History] and the [Next Item] on a scale from 1 to 7 (1: Very Weak - No connection, 2: Weak - Poorly justified, 3: Slightly Weak - Minimal relevance, 4: Neutral - Ambiguous or marginally related, 5: Slightly Strong- Generally aligned, 6: Strong: Well aligned and logical, 7: Very Strong - Highly coherent and contextually perfect), and insert the score within the `<rating></rating>`tag. Be critical, not generous.
>
> For example, write it like this:
> `<think>`your reasoning process `</think>` $\leftarrow r_c$ or $r_h$ or $r_e$
> `<rating>`the degree of the logical coherence `</rating>` $\leftarrow s_c$ or $s_h$ or $s_e$

**Step2: Human Evaluation Protocol**

After generating the recommendation rationales $r$ and their corresponding coherence scores $s$ from the LLM, we conducted a statistical validation to assess whether human evaluators make similar judgments regarding the quality of $r$ as those reflected in $s$. If the agreement is not statistically significant, we return to Step 1 and revise the generation process. Specifically, seven domain experts[9] in the field of recommender systems evaluated approximately 400 samples from the LLM-generated dataset by assigning a coherence score on a 1–7 Likert scale, following the same instructions provided to the LLM (See the above prompt). The scores assigned by the experts were subsequently aggregated to obtain the mean human-evaluated coherence score per sample. We then performed a Spearman's rank correlation test between the LLM-generated coherence scores and the human ratings, obtaining a statistically significant correlation coefficient of $\rho = 0.71$, $p < 0.0001$. Additionally, the Quadratic Weighted Kappa (QWK) score was 0.63, indicating substantial agreement between the model-generated scores and human evaluations. To further visualize the distribution of agreement between human and LLM assessments, we present the confusion matrix in Fig. 8. **This result indicates a substantial agreement between LLM assessments and human intuition, supporting the reliability of the coherence score as a meaningful training signal.**

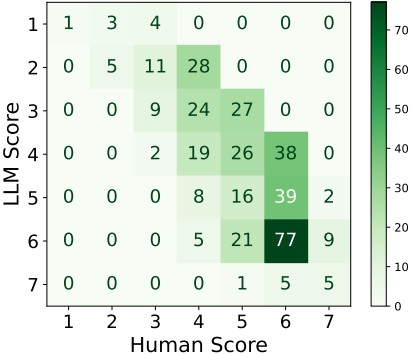

Figure 8: Confusion matrix between LLM and Human Coherence Score

**Step3: Data Filtering Criteria** Subsequently, for training data construction, we filtered out all instances where the two rejected items received identical coherence scores. As a result, our final dataset consists exclusively of triplets in the form of (chosen, hard rejected, easy rejected), where the hard rejected item is guaranteed to have a higher coherence score than the easy rejected item. This filtering step preserved approximately 70% of the original data instances.

Table 5: Rationale Quality

| History Length | Rationale Quality | | |
| --- | --- | --- | --- |
| | count | mean | std |
| 1–2 | 998 | **2.20** | 0.81 |
| 3–7 | 538 | 2.68 | 0.82 |
| 8–11 | 142 | 2.70 | 0.81 |
| 12+ | 220 | 2.76 | 0.71 |

We additionally conducted an experiment in which we generated rationales for users with only 1–2 interactions (sampled from approximately 2,000 instances in the raw Amazon review data) and evaluated their quality using the same 0–3 protocol described in Section 4.4. Histories of length 1–2 produced substantially lower-quality rationales, as shown in Table 5.

---

[9]A total of seven domain experts participated in the evaluation: one service marketer (female, age 34); two service engineers (male, ages 41 and 29); and four data scientists (male, ages 37, 33, 31, and 30). This group reflects diverse professional roles spanning product, engineering, and modeling. All annotators scored a shared subset of items, enabling measurement of inter-annotator agreement. For cases where annotators' scores differed by more than 4 points on the 7-point scale, a consensus procedure was conducted to resolve discrepancies.

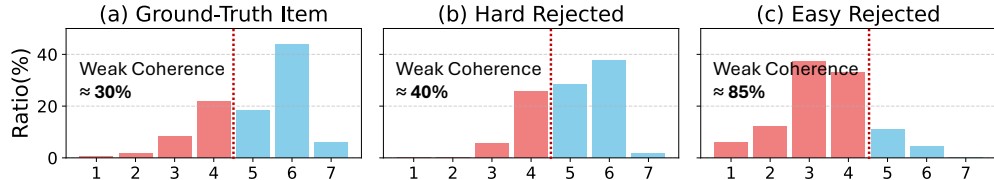

Figure 9: Distribution of coherence scores (LLM-as-a-Judge) for *ground-truth* (Chosen), *hard rejected*, and *easy rejected* items in the Amazon dataset, computed from the relationship between each item and the user's interaction history (1–7 scale, higher = more logical). *Chosen* items are generally distributed in the higher score range, while *hard* and *easy rejected* items cluster in lower ranges.

## E    PROMPT TEMPLATE FOR C-APO

As described in Appendix D, after constructing the training dataset, we first performed supervised fine-tuning, followed by reinforcement learning based on the C-APO framework. The training prompt includes the user's purchase/view history, a list of candidate items, and an instruction. The model is tasked with generating a recommended item—either a chosen, hard rejected, or easy rejected item—accompanied by a rationale for each. For comparison, our experiments configure DPO to use a single randomly selected negative item (either hard- or easy rejected), whereas S-DPO—designed to model multiple negative items—leverages the entire triplet structure comprising the chosen, hard rejected, and easy rejected items, as does our method. An illustrative example of such a prompt is presented below.

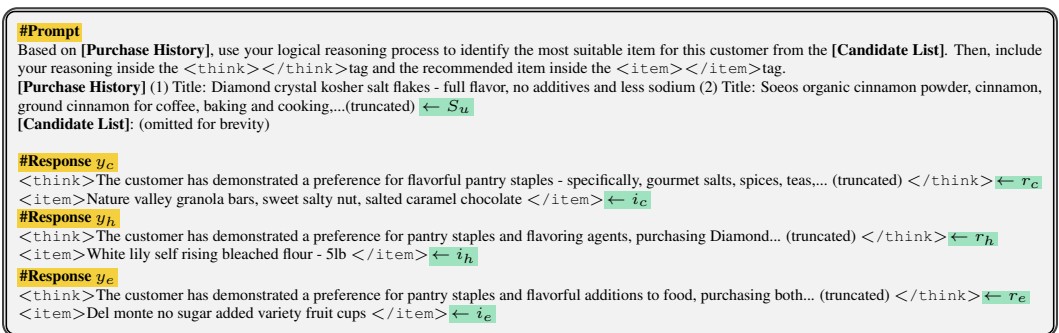

As mentioned earlier, this prompt is shared across all models—DPO (one rejected item), S-DPO (two rejected items, and the PL model (two rejected items)—differing only in the number of responses used.

## F    RATIONALE EVALUATION DETAILS

We describe our evaluation procedure for LLM-generated rationales, as introduced in Section 4.4. First, we tasked the SOTA LLM itself (i.e., `gpt-4o-series`) with scoring the coherence of each generated rationale according to a predefined rubric. Importantly, the LLM used for this scoring step is distinct from the one employed to generate the rationales during dataset construction, ensuring an independent evaluation. To assess the reliability of these automatic evaluations, a subset of samples was independently annotated by human judges, who followed the same evaluation criteria provided to the LLM. We then conducted statistical tests to examine the degree of alignment between LLM- and human-assigned scores, and found a statistically significant positive correlation—supporting the validity of the LLM-based rationale assessment.

**Rationale Evaluation Prompt**    We assess the plausibility of the generated recommendations by prompting a state-of-the-art LLM to evaluate the rationales on a four-point scale, using the following prompt.

***Prompt for the Rationales Evaluation***

**#Instruction**

Please act as a neutral evaluator and assess the AI assistant's recommendation and its rationales based on the user's purchase history, the predicted item, and the ground truth item. Use the following four-level scale to assign your rating:

**RATING-0**: Incorrect recommendation — The assistant fails to recommend the correct item.

**RATING-1**: Correct recommendation, poor explanation — While the assistant recommends the correct item, the explanation is missing, vague, off-topic, or includes hallucinated content that doesn't align with the actual context.

**RATING-2**: Correct recommendation, reasonable explanation — The assistant offers a recommendation that fits the user's profile and provides a logically coherent explanation. However, the explanation may still lack depth, clarity, or persuasive detail.

**RATING-3**: Correct recommendation, strong explanation — The assistant not only recommends the right item but also presents a clear and insightful explanation, referencing the user's behavior patterns and showing how they relate to the suggested item.

Please submit your evaluation using the format <eval>RATING-n </eval>, for instance: <eval>RATING-2 </eval>. Avoid letting the explanation length affect your judgment. Aim for an objective assessment.

**#Known Information**

**[Purchase history]** (1) Title: Fansing costume jewelry fathers day gift punk screw stud earrings (2) Title: Iblue jewelry stainless steel 3 pairs mens black gold silver stud earrings set,...(truncated)

**[Recommended Item]** Jstyle stainless steel mens womens stud earring hoop earrings for men 7 pairs

**[Ground-Truth Item]** Jstyle stainless steel mens womens stud earring hoop earrings for men 7 pairs

**[Rationale]** The customer previously purchased two sets of stainless steel stud earrings - one with a punk aesthetic and... (truncated)

**Agreement Analysis between LLM and Human Ratings** To validate the LLM-derived ratings of rationale quality, we asked two domain experts to evaluate about 250 samples drawn from a total of 1,500 instances (Section 4.4). To assess the agreement between LLM-generated and human-assigned scores, we report Quadratic Weighted Kappa (QWK = 0.75, $p < 0.0001$) and Spearman's rank correlation ($\rho = 0.63$, $p < 0.0001$). Both metrics indicate strong agreement: the QWK score reflects substantial concordance while accounting for the ordinal scale and penalizing large discrepancies, whereas Spearman's $\rho$ captures the consistency in ranking and similarly demonstrates high alignment. The results indicate a moderate level of agreement in absolute scoring and a fairly consistent ranking of item quality, support-

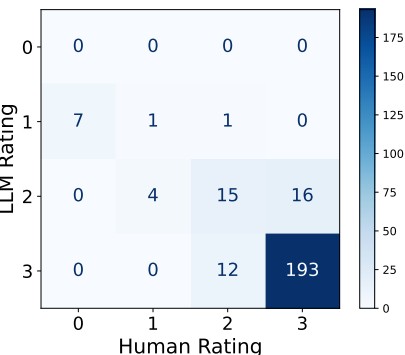

Figure 10: Confusion matrix between LLM and Human Rationale Rating

ing the validity of the LLM-based ratings and their alignment with human evaluations. To further examine how ratings differ across score levels, we visualize the confusion matrix between human and LLM scores (Fig.10). Most predictions are concentrated along the diagonal, indicating that the LLM generally assigns scores close to human judgments, with only a small proportion of large discrepancies.

**Case Studies of Rationales** To better illustrate the qualitative differences in recommendation generation across models, we provide representative examples in Fig.11 and Fig.12, comparing our method against several competitive baselines.

Our model consistently generates rationales that are not only more fluent but also more logically grounded in the user's actual purchase history. For example, in Case 1 (Fig.11), our method identifies the user's strong preference for women's clothing—particularly dresses and skirts—and recommends a women's cropped cardigan that naturally complements those items. In contrast, Rec-SAVER infers a general preference for practicality and suggests men's jeans, overlooking the user's clear pattern of purchasing feminine apparel. S-DPO, on the other hand, focuses on the user's appreciation for expressive and practical items but recommends a humorous graphic t-shirt, which mismatches the more formal tone of previous purchases. These comparisons demonstrate that our model offers recommendations and rationales that are more coherent, contextually appropriate, and better aligned with the user's underlying preferences.

Another representative example is illustrated in Fig. 12. Our model identifies a clear pattern of preference for durable, outdoor-ready products, such as Casio G-Shock watches and Carhartt work pants, and recommends Timberland waterproof hiking boots that naturally extend this theme through similar emphasis on ruggedness and functionality. In contrast, Rec-SAVER captures a general interest in utility but recommends moisture-wicking socks, which, while practical, lack the salience and robustness of prior purchases. S-DPO focuses on the user's appreciation for high-performance accessories but similarly suggests socks, overlooking the broader pattern of activewear and heavy-

duty gear. These comparisons highlight that our method yields recommendations and rationales that are more coherent, contextually grounded, and better aligned with user preferences.

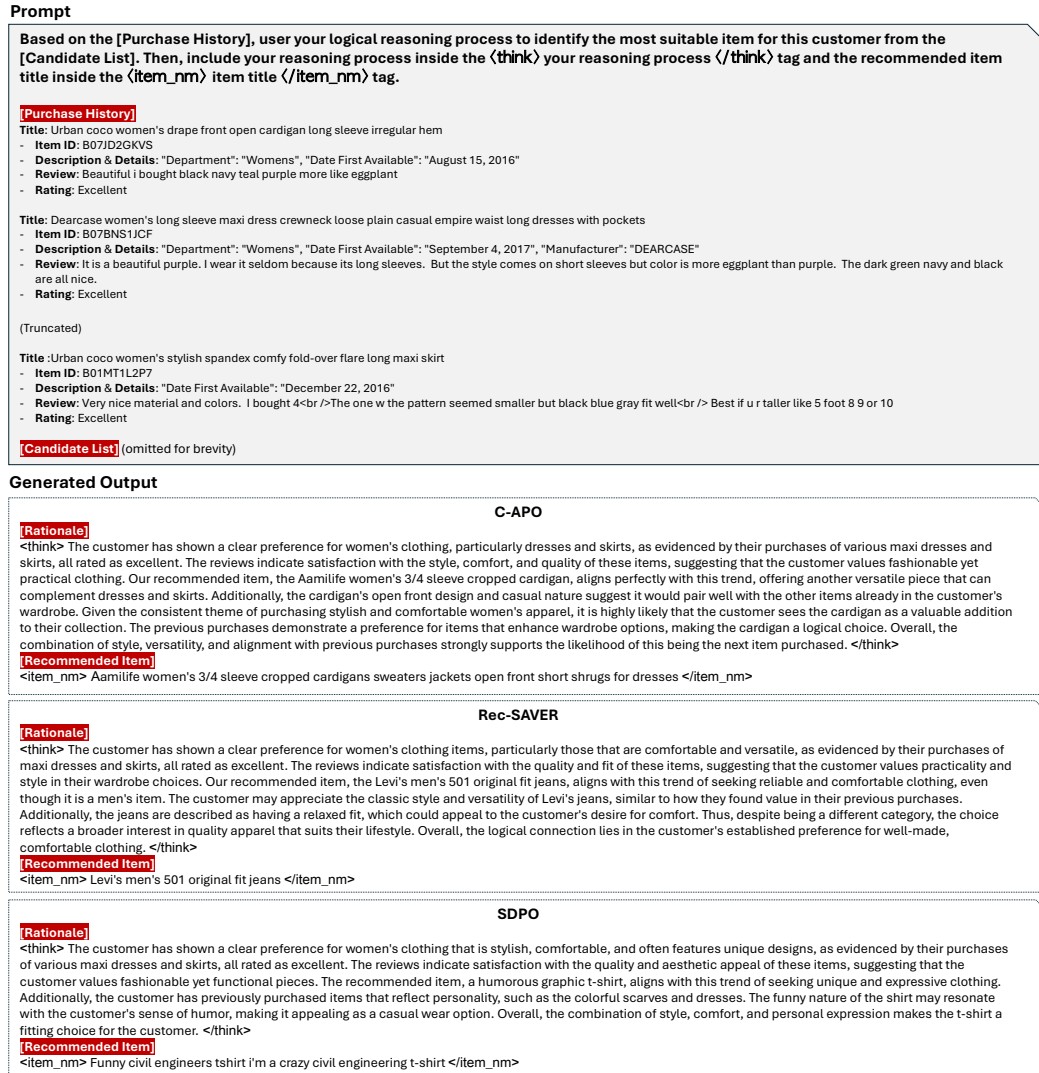

Figure 11: Different rationale qualities across models for the same prompt (Case 1)

## G  SERENDIPITY ANALYSIS

Because C-APO increases the likelihood of history-consistent items, one concern is that it may over-anchor on past behaviors and reduce exploratory recommendations. To assess this, we analyze serendipity using the standard metric $\text{Serendipity}(u) = \frac{|R_u \cap T_u \setminus P|}{|R_u|}$, where $R_u$ are recommendations, $T_u$ true interactions, and $P$ globally popular items. Empirically, C-APO achieves a serendipity score of 0.30 versus 0.25 for the SFT-only model, indicating that it recommends more relevant yet non-popular items rather than collapsing into an echo-chamber pattern. Moreover, as shown in Fig. 6, moderate C-APO alignment (e.g., $\beta \leq 1$) provides a favorable balance between accuracy and exploratory diversity.

**Prompt**

Based on the [Purchase History], user your logical reasoning process to identify the most suitable item for this customer from the [Candidate List]. Then, include your reasoning process inside the ⟨think⟩ your reasoning process ⟨/think⟩ tag and the recommended item title inside the ⟨item_nm⟩ item title ⟨/item_nm⟩ tag.

[Purchase History]
**Title**: Casio g-shock gw-7900b-1er men's watch, black
- **Item ID**: B0039YOIH0
- **Description & Details**: With the never ending pursuit of toughness and technology, G-Shock introduces a solar atomic addition to the tough and rugged G-Rescue series. ...
- **Review**: Loved the watch. I have owned it for about 10 months. Didn't wear it every day, but after about 3 months of constant wear, ...
- **Rating**: Excellent

**Title**: Casio men's gw-9400-1cr master of g stainless steel solar watch
- **Item ID**: B00GFYYKLA
- **Description & Details**: Casio Rangeman GW9400-1 AS MB6 Twin Sensor Wristwatch,Casio, ...
- **Review**: Awesome watch!
- **Rating**: Excellent

(Truncated)

**Title** :Carhartt men's loose fit firm duck double-front utility work pant
- **Item ID**: B002G9U3WS
- **Description & Details**: Double up on durability with carhartt's double-front work dungaree in 12-ounce, firm-hand, 100-percent ring-spun cotton duck, it features chap-style, double-front panels as well as double knees with cleanout bottoms to accommodate knee pads, ...
- **Review**: These pants are awesome! Very tough and lots of pockets to carry my gear. I do rigorous field work and have worn these pants for 20 years.
- **Rating**: Excellent

[Candidate List] (omitted for brevity)

**Generated Output**

**C-APO**

[Rationale]
<think> The customer has previously purchased multiple Casio watches, specifically models known for their durability and functionality, such as the G-Shock and Rangeman watches, both rated as excellent. These watches indicate a preference for rugged, reliable products that can withstand harsh conditions, suggesting that the customer values quality and performance in their purchases. Our recommended item, the Timberland men's white ledge mid waterproof hiking boot, aligns perfectly with this preference, as it emphasizes durability and waterproofing, essential traits appreciated in outdoor activities. Additionally, the customer has shown interest in outdoor and functional clothing, as evidenced by their purchase of the Carhartt work pants, which also highlight practicality and resilience. The Timberland boots complement this trend by offering a solid construction suited for various terrains, reinforcing the customer's inclination towards durable footwear. Therefore, the logical connection between the customer's past purchases and the next item indicates a strong likelihood of purchasing the Timberland hiking boots. </think>
[Recommended Item]
<item_nm> Timberland men's white ledge mid waterproof hiking boot </item_nm>

**Rec-SAVER**

[Rationale]
<think> The customer has shown a strong preference for durable and functional products, particularly those designed for outdoor activities, as evidenced by their purchases of Casio G-Shock watches and Carhartt work pants, both rated as excellent. The Carhartt work pants were praised for their toughness and practicality, aligning with the customer's active lifestyle. Additionally, the customer has purchased multiple Casio watches, indicating a consistent interest in reliable and robust brands known for their quality and functionality. The Dickies men's dri-tech moisture control crew socks multipack complements this trend by offering moisture management and comfort, essential features for someone engaged in physical activities. Given the customer's previous positive experiences with high-performance apparel and footwear, it is logical that they would choose another item that enhances their overall performance during work or outdoor settings. Therefore, the decision to purchase the Dickies socks aligns perfectly with their established preferences for quality and functionality in their purchases. </think>
[Recommended Item]
<item_nm> Dickies men's dri-tech moisture control crew socks multipack </item_nm>

**SDPO**

[Rationale]
<think> The customer has previously purchased multiple Casio G-Shock watches, indicating a strong preference for durable, rugged, and technologically advanced products. Each of these watches received excellent ratings and positive reviews, highlighting their toughness, functionality, and ease of readability. Our recommended item, the Dickies men's dri-tech moisture control crew socks multipack, aligns with the customer's practical approach to purchasing high-quality, functional items that withstand daily wear and tear, similar to the watches they favor. Additionally, the socks are designed for comfort and performance, which resonates with the customer's appreciation for quality materials and functionality. Given the customer's consistent choice of durable, reliable brands, the decision to purchase another pair of high-performance socks from a reputable brand like Dickies is a logical continuation of their buying pattern. </think>
[Recommended Item]
<item_nm> Dickies men's dri-tech moisture control crew socks multipack </item_nm>

Figure 12: Different rationale qualities across models for the same prompt (Case 2)

# H EXPERIMENTS ON OTHER DOMAINS

## H.1 COLD-START AND ZERO-SHOT DOMAIN TRANSFER

LLM-based recommender systems (LLM-Rec) have been shown to exhibit strong robustness in cold-start scenarios, often outperforming traditional collaborative filtering recommenders (Lei et al., 2024). Because LLMs encode broad semantic knowledge, they can also provide reasonable predictions for domains that were never observed during training—a capability that CF-based models typically lack. To evaluate this aspect, we examined the zero-shot domain transfer ability of our approach (Table 6).

In this experiment, the CF-Rec baselines were trained *directly* on the target (unseen) domains, whereas our model was evaluated without any domain-specific supervision. We considered two representative target domains: *Amazon Beauty* and *CDs & Vinyl*. For our model, we selected semantically related source domains for training: *Amazon Fashion* for *Amazon Beauty*, and *Musical Instruments* for *CDs & Vinyl*. The results show that, despite not being trained on these target domains, our model achieves performance comparable to CF-Rec

Table 6: Performance Comparison on Unseen Domains

| Domain | Amazon Beauty | | | CDs and Vinyl | | |
|---|---|---|---|---|---|---|
| **Models** | HR@1 | HR@5 | N@5 | HR@1 | HR@5 | N@5 |
| SASRec | 2.90 | 16.12 | 9.18 | 3.26 | 16.34 | 9.61 |
| BERT4Rec | 3.26 | 15.04 | 9.01 | 3.14 | 14.96 | 8.89 |
| GRU4Rec | 3.26 | 17.57 | 10.51 | 4.29 | 24.00 | 13.73 |
| FOSSIL | 1.81 | 13.95 | 7.84 | 3.72 | 15.74 | 9.59 |
| NextItNet | 2.72 | 14.67 | 8.45 | 3.68 | 15.65 | 9.57 |
| TransRec | 3.62 | 17.21 | 10.33 | 2.93 | 14.90 | 8.69 |
| SINE | 3.80 | 15.04 | 9.26 | 4.64 | 18.04 | 11.23 |
| Rec-SAVER | 8.56 | 31.51 | 19.69 | **10.16** | 27.84 | 19.40 |
| **Ours** (Unseen) | **19.49** | **40.62** | **30.81** | 10.08 | **28.44** | **19.66** |

models trained explicitly on the unseen domains. This demonstrates that C-APO effectively captures user behavioral patterns and item semantics in a manner that generalizes well across domains, highlighting its robustness in cold-start and unseen-domain scenarios.

## H.2 EXPERIMENTS ON NON-AMAZON DOMAINS (MOVIELENS)

To verify that C-APO is not limited to Amazon-style review data, we additionally evaluated the model on the MovieLens dataset, which differs substantially in structure despite also being text-based.

Table 7: Performance Comparison on the MovieLens Dataset

| Model | BERT4Rec | GRU4Rec | NextItNet | SINE | STAMP | Ours |
|---|---|---|---|---|---|---|
| HR@1 | 2.26 | 2.51 | 2.25 | 1.93 | 2.08 | **2.81** |
| HR@5 | 12.84 | 13.43 | 10.41 | 11.60 | 9.86 | **13.13** |
| NDCG@5 | 7.38 | 7.86 | 6.24 | 6.61 | 5.88 | **8.10** |

Unlike Amazon reviews that contain rich free-form descriptions, MovieLens provides only short user-generated tags, making it a fundamentally different type of textual signal.

Even under this distinct data structure, C-APO consistently outperformed all baseline models, reflecting the same performance trends observed in the five Amazon domains (Table 7). These results demonstrate that the core principles of our approach—CP-based reasoning and conflict-aware optimization—generalize beyond Amazon datasets and remain effective in structurally different domains.

## I GRADIENT DERIVATION FOR $\mathcal{L}_{\text{C-APO}}$

**Setup and notation.** Let $g_k := g(x_u, y_k)$ be the model reward for item $y_k$ given user context $x_u$, and define

$$\Delta g_{h,c} = g_h - g_c, \quad \Delta g_{e,c} = g_e - g_c, \quad \Delta g_{e,h} = g_e - g_h, \quad \nabla_k := \nabla_\theta g(x_u, y_k).$$

The C-APO loss in Eq. 6 can be written as

$$\mathcal{L}_{\text{C-APO}} = -\mathbb{E}\Big[ \log \sigma\Big( -\log\big( e^{w_{c,h}\Delta g_{h,c}} + e^{w_{c,e}\Delta g_{e,c}} \big) \Big) + \log \sigma\Big( -\log\big( e^{w_{h,e}\Delta g_{e,h}} \big) \Big) \Big], \quad (10)$$

where $\sigma(\cdot)$ is the sigmoid function and $w_{c,h}, w_{c,e}, w_{h,e} > 0$ are the conflict-aware adaptive weights. For clarity in gradient analysis, the trainable layers of the conflict-aware adaptive weight module were excluded from differentiation.

**Useful identity.** For $z = z(\theta)$, we will use

$$\frac{\partial}{\partial \theta} \log \sigma(-z) = \sigma(z) \cdot \Big( -\frac{\partial z}{\partial \theta} \Big). \quad (11)$$

Therefore, with the leading negative sign in Eq.10, each term contributes a factor $+\sigma(z) \cdot \partial_\theta z$ to the gradient.

**First term (RP aligned with CP).** Let

$$z_1 := \log\big( e^{w_{c,h}\Delta g_{h,c}} + e^{w_{c,e}\Delta g_{e,c}} \big) = \log S, \quad S := e^{w_{c,h}\Delta g_{h,c}} + e^{w_{c,e}\Delta g_{e,c}}.$$

By the chain rule,

$$\partial_\theta z_1 = \frac{1}{S} \partial_\theta S = \frac{1}{S} \Big( e^{w_{c,h}\Delta g_{h,c}} \cdot w_{c,h} \partial_\theta \Delta g_{h,c} + e^{w_{c,e}\Delta g_{e,c}} \cdot w_{c,e} \partial_\theta \Delta g_{e,c} \Big).$$

Since $\Delta g_{h,c} = g_h - g_c$ and $\Delta g_{e,c} = g_e - g_c$,

$$\partial_\theta \Delta g_{h,c} = \nabla_h - \nabla_c, \qquad \partial_\theta \Delta g_{e,c} = \nabla_e - \nabla_c.$$

Hence

$$\partial_\theta z_1 = \frac{w_{c,h} e^{w_{c,h}\Delta g_{h,c}} (\nabla_h - \nabla_c) + w_{c,e} e^{w_{c,e}\Delta g_{e,c}} (\nabla_e - \nabla_c)}{e^{w_{c,h}\Delta g_{h,c}} + e^{w_{c,e}\Delta g_{e,c}}}. \quad (12)$$

Using Eq. 11 with the outer minus sign in Eq. 10, the first term contributes

$$\boxed{\sigma(z_1) \cdot \partial_\theta z_1 = \sigma\Big( \log\big( e^{w_{c,h}\Delta g_{h,c}} + e^{w_{c,e}\Delta g_{e,c}} \big) \Big) \frac{w_{c,h} e^{w_{c,h}\Delta g_{h,c}} (\nabla_h - \nabla_c) + w_{c,e} e^{w_{c,e}\Delta g_{e,c}} (\nabla_e - \nabla_c)}{e^{w_{c,h}\Delta g_{h,c}} + e^{w_{c,e}\Delta g_{e,c}}}.}$$
$$(13)$$

**Second term (CP).** Since $\log\left(e^{w_{h,e}\Delta g_{e,h}}\right) = w_{h,e}\Delta g_{e,h}$, define

$$z_2 := w_{h,e}\Delta g_{e,h} = w_{h,e}(g_e - g_h).$$

Then

$$\partial_\theta z_2 = w_{h,e}\,\partial_\theta \Delta g_{e,h} = w_{h,e}(\nabla_e - \nabla_h). \tag{14}$$

By the same identity Eq. 11 (with the leading minus in the loss), the second term contributes

$$\boxed{\sigma(z_2)\cdot\partial_\theta z_2 = \sigma\!\left(w_{h,e}\Delta g_{e,h}\right) w_{h,e}\,(\nabla_e - \nabla_h).} \tag{15}$$

**Final gradient.** Combining Eq. 13 and Eq.15 inside the expectation yields

$$\nabla_\theta \mathcal{L}_{\text{C-APO}}$$

$$= \mathbb{E}\!\left[\sigma\!\Big(\log\big(e^{w_{c,h}\Delta g_{h,c}} + e^{w_{c,e}\Delta g_{e,c}}\big)\Big)\frac{w_{c,h}e^{w_{c,h}\Delta g_{h,c}}(\nabla_h - \nabla_c) + w_{c,e}e^{w_{c,e}\Delta g_{e,c}}(\nabla_e - \nabla_c)}{e^{w_{c,h}\Delta g_{h,c}} + e^{w_{c,e}\Delta g_{e,c}}} \ + \ \sigma\!\big(w_{h,e}\Delta g_{e,h}\big)w_{h,e}\,(\nabla_e - \nabla_h)\right],$$

$$= -\mathbb{E}\!\left[\sigma\!\Big(\log\big(e^{w_{c,h}\Delta g_{h,c}} + e^{w_{c,e}\Delta g_{e,c}}\big)\Big)\frac{w_{c,h}e^{w_{c,h}\Delta g_{h,c}}(\nabla_c - \nabla_h) + w_{c,e}e^{w_{c,e}\Delta g_{e,c}}(\nabla_c - \nabla_e)}{e^{w_{c,h}\Delta g_{h,c}} + e^{w_{c,e}\Delta g_{e,c}}} \ + \ \sigma\!\big(w_{h,e}\Delta g_{e,h}\big)w_{h,e}\,(\nabla_h - \nabla_e)\right], \tag{16}$$

which corresponds exactly to Eq. 8 presented in the main paper.

## J  PROBABILISTIC INTERPRETATION OF THE PAIRWISE WEIGHTS

We provide a generative interpretation for the conflict-aware adaptive weights $w_{i,j}$ used in our objective. Let the latent utility of item $i \in \{c, h, e\}$ be

$$U_i \sim \mathcal{N}(\mu_i, \tilde{\sigma}_i^2),$$

where $(\mu_i, \tilde{\sigma}_i)$ are (calibrated) parameters inferred from the text-encoder features of the user–item–rationale tuple. Then, for any pair $(i, j)$,

$$w_{i,j} \ \equiv \ \Pr[U_i > U_j] \ = \ \Pr[U_i - U_j > 0] \ = \ \Phi\!\left(\frac{\mu_i - \mu_j}{\sqrt{\tilde{\sigma}_i^2 + \tilde{\sigma}_j^2}}\right),$$

where $\Phi(\cdot)$ denotes the standard Gaussian CDF. Therefore, the weight $w_{i,j}$ is the *pairwise win probability* of $i$ over $j$ under a Thurstone–Mosteller model of noisy utilities (Thurstone, 2017; Mosteller, 1951). The Thurstone–Mosteller model is a classical instance of the random utility framework, in which the utility of each alternative in a pairwise comparison is modeled as a latent score perturbed by Gaussian noise. This model has been widely adopted as a foundational approach for modeling pairwise preferences and ranking data across various domains, including social sciences, psychology, and recommender system.

Substituting $w_{i,j}$ into our loss yields a *probability-weighted* variant of the PL/DPO family in which pairwise gaps $(g_i - g_j)$ are modulated by their estimated reliability.

**Consequence.** When $(\mu_i - \mu_j)$ is large relative to the joint uncertainty $\sqrt{\tilde{\sigma}_i^2 + \tilde{\sigma}_j^2}$, the model assigns $w_{i,j} \approx 1$, amplifying the update in the direction that promotes $i \succ j$; when the pair is uncertain or contradictory, $w_{i,j} \approx \frac{1}{2}$, attenuating the update; and when $(\mu_i - \mu_j)$ is negative, $w_{i,j}$ approaches 0, thereby weakening the update that favors $i \succ j$.

## K  BOUNDARY CASES AND REDUCTIONS

We collect simple but instructive reductions of our objective.

**Lemma 1** (Reduction to Plackett-Luce (PL) model). *If $w_{i,j} \equiv 1$ for all pairs, $\mathcal{L}_{\text{C-APO}}$ (Eq. 6) reduces exactly to the PL objective that jointly enforces $y_c \succ y_h \succ y_e$ via a two-stage factorization.*

**Lemma 2** (RP-only limit). *If the CP comparison is disabled by setting $w_{h,e} = 0$ (or by dropping the CP term) and $w_{c,h}, w_{c,e} = 1$, $\mathcal{L}_{\text{C-APO}}$ reduces to the S-DPO objective (Chen et al., 2024) that promotes $y_c$ above $\{y_h, y_e\}$.*

**Lemma 3** (Temperature rescaling). *If $w_{i,j} \equiv \lambda \in (0, \infty)$ is a constant, then $\mathcal{L}_{\text{C-APO}}$ is equivalent to applying a temperature $1/\lambda$ to all pairwise gaps inside the log-sum-exp. This does not change the target ordering but rescales the sharpness of the updates.*

**Implication.** These reductions clarify that our method strictly generalizes standard PL/DPO-style training: it recovers PL when uncertainty is ignored ($w \equiv 1$), focuses on RP when CP is omitted ($w_{h,e} = 0$), and smoothly interpolates update strength via a temperature view when $w$ is constant.

## L    REFINED COMPLEXITY STATEMENTS

Let $C_{\text{LLM}}$ denote the cost of one forward/backward pass of the backbone LLM for a single $(x, y)$, $B$ the batch size, and $M = 1 + R$ the number of candidate responses (one chosen, $R$ rejected; here $R=2$).

**Proposition 1** (Training cost). *C-APO evaluates $M$ responses and combines them via a two-stage PL factorization, yielding $\mathcal{O}(B\,M\,C_{\text{LLM}})$ per step. For DPO with a single pair $(c, -)$, the cost is $\mathcal{O}(2B\,C_{\text{LLM}})$. If the calibrator uses a lightweight encoder with cost $C_{enc} \ll C_{\text{LLM}}$, its contribution is negligible to leading order.*

**Proposition 2** (Inference cost). *At inference, scoring $M$ candidates costs $\mathcal{O}(M\,C_{\text{LLM}})$; aggregation (e.g., softmax/log-sum-exp over $M$) is lower order. Thus latency is dominated by backbone evaluation and scales linearly in $M$ for a fixed input length.*

C-APO matches the leading-order complexity of PL/DPO-style training while adding only a lightweight weighting step; practical latency is governed by the backbone, not by the calibrator.

## M    DISCUSSION OF SCALING-LAW

We conducted a comprehensive evaluation using backbone models with parameter sizes of 1B (Gemma-3-1B-it), 4B (Gemma-3-4B-it), and 12B (Gemma-3-12B-it). As illustrated in Fig. 13, model performance exhibited a consistent upward trend with increased model size across both the *Grocery and Gourmet Food* and *Clothing Shoes and Jewelry* domains. All three evaluation metrics—HR@1, HR@5, and NDCG@5—improved monotonically as the number of model parameters increased, reflecting a clear scaling effect in recommendation performance.

Despite these gains, increasing model size also leads to higher inference latency, which can be a limiting factor in real-time deployment scenarios. To strike a balance between performance and responsiveness, we adopt the 4B parameter model for integration into our production environment.

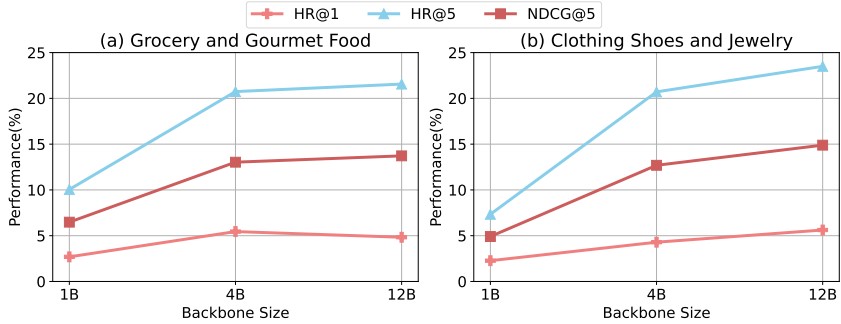

Figure 13:  Performance of C-APO improves monotonically with larger LLM backbones (Gemma-3-1B/4B/12B), confirming a scaling-law trend across two domains.

