^-)\}$, with $x_u$, $y_c$, and $y^-$ denoting the input prompt, chosen item, and rejected item, respectively, preference can be formulated by the Bradley-Terry model as $p(y_c \succ y^- | x_u) = \sigma(g(x_u, y_c) - g(x_u, y^-))$. DPO formulates the preference likelihood as the following objective:

$$\mathcal{L}_{\text{DPO}}(\pi_\theta; \pi_{\text{ref}}) = -\mathbb{E}_{(x_u, y_c, y^-) \sim \mathcal{D}} \left[ \log \sigma \left( \beta \log \frac{\pi_\theta(y_c|x_u)}{\pi_{\text{ref}}(y_c|x_u)} - \beta \log \frac{\pi_\theta(y^-|x_u)}{\pi_{\text{ref}}(y^-|x_u)} \right) \right]. \quad (1)$$

However, existing applications of DPO in recommender systems primarily rely on RP, under the assumption that the chosen item is preferred over the rejected one. To overcome this limitation, we propose **C-APO**, a novel framework that integrates both RP, which captures user–item interaction signals, and CP, which models the reasoning behind choices—thereby improving recommendation performance and enhancing the persuasiveness of generated rationales.

### 3.2 DERIVATION OF C-APO

As mentioned in Section 2.2, we are given a dataset of user preference triplets $\mathcal{D} = \{(x, y_c, y_h, y_e)\}$. To align LLM-Rec with triplet preference, we first derive the preference distribution.

**Plackett-Luce (PL) model** For a prompt $x_u$ and three candidate responses $\{y_c, y_h, y_e\}$, the Plackett–Luce (PL) probability of a permutation $\tau$ is as follows:

$$p(\tau \mid y_c, y_h, y_e, x_u) = \prod_{j=1}^{3} \frac{e^{\left(g_\theta(x_u, y_{\tau(j)})\right)}}{\sum_{l=j}^{3} e^{\left(g_\theta(x_u, y_{\tau(l)})\right)}}, \quad (2)$$

where $e^x$ denotes the exponential function, with the shorthand $e$ used in place of $\exp(\cdot)$ due to space constraints, and the implicit reward is given by $g_\theta(x, y) = \beta \log \frac{\pi_\theta(y|x)}{\pi_{\text{ref}}(y|x)}$. An example of the prompt used for this task is provided in Appendix E, which is also used in our C-APO framework.

For the desired full ranking $y_c \succ y_h \succ y_e$, let $\tau^\star = (y_c, y_h, y_e)$. Within $\tau^\star$, RP is formalized as $y_c \succ y_h$ and $y_c \succ y_e$, whereas CP is formalized as $y_h \succ y_e$. With shorthand $g_c = g_\theta(x_u, y_c)$, $g_h = g_\theta(x_u, y_h)$, $g_e = g_\theta(x_u, y_e)$, Eq. 2 can be rewritten as:

$$p^*(y_c \succ y_h \succ y_e \mid x_u) = p^*(\tau^\star \mid y_c, y_h, y_e, x_u) = \frac{e^{g_c}}{e^{g_c} + e^{g_h} + e^{g_e}} \cdot \frac{e^{g_h}}{e^{g_h} + e^{g_e}}. \quad (3)$$

We maximize the log–likelihood of the complete ordering; the loss function is therefore the negative log-probability:

$$\mathcal{L}_{PL} = -\mathbb{E}_{(x_u, y_c, y_h, y_e) \sim \mathcal{D}}\big[\log p^*(y_c \succ y_h \succ y_e \mid x_u)\big]. \quad (4)$$

Substituting Eq. 3 into Eq. 4 and simplifying yields:

$$\mathcal{L}_{PL} = -\mathbb{E}_{(x_u, y_c, y_h, y_e) \sim \mathcal{D}}\Big[\underbrace{\log \sigma\Big(-\log\Big(e^{(g_h - g_c)} + e^{(g_e - g_c)}\Big)\Big)}_{\text{(1) Revealed Preference (RP)}} + \underbrace{\log \sigma\Big(-\log\Big(e^{(g_e - g_h)}\Big)\Big)}_{\text{(2) Coherent Preference (CP)}}\Big]. \quad (5)$$

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

[**Candidate List**]: (omitted for brevity)

**#Response** $y_c$
<think>The customer has demonstrated a preference for flavorful pantry staples - specifically, gourmet salts, spices, teas,... (truncated) </think>$\leftarrow r_c$
<item>Nature valley granola bars, sweet salty nut, salted caramel chocolate </item>$\leftarrow i_c$
**#Response** $y_h$
<think>The customer has demonstrated a preference for pantry staples and flavoring agents, purchasing Diamond... (truncated) </think>$\leftarrow r_h$
<item>White lily self rising bleached flour - 5lb </item>$\leftarrow i_h$
**#Response** $y_e$
<think>The customer has demonstrated a preference for pantry staples and flavorful additions to food, purchasing both... (truncated) </think>$\leftarrow r_e$
<item>Del monte no sugar added variety fruit cups </item>$\leftarrow i_e$
---

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

- **Item ID**: B07JD2GKVS
- **Description** & Details: "Department": "Womens", "Date First Available": "August 15, 2016"
- **Review**: Beautiful i bought black navy teal purple more like eggplant
- **Rating**: Excellent

**Title**: Dearcase women's long sleeve maxi dress crewneck loose plain casual empire waist long dresses with pockets
- **Item ID**: B07BNS1JCF
- **Description** & Details: "Department": "Womens", "Date First Available": "September 4, 2017", "Manufacturer": "DEARCASE"
- **Review**: It is a beautiful purple. I wear it seldom because its long sleeves. But the style comes on short sleeves but color is more eggplant than purple. The dark green navy and black are all nice.
- **Rating**: Excellent

(Truncated)

**Title** :Urban coco women's stylish spandex comfy fold-over flare long maxi skirt
- **Item ID**: B01MT1L2P7
- **Description** & Details: "Date First Available": "December 22, 2016"
- **Review**: Very nice material and colors. I bought 4
The one w the pattern seemed smaller but black blue gray fit well
 Best if u r taller like 5 foot 8 9 or 10
- **Rating**: Excellent

[Candidate List] (omitted for brevity)

**Generated Output**

**C-APO**

[Rationale]
<think> The customer has shown a clear preference for women's clothing, particularly dresses and skirts, as evidenced by their purchases of various maxi dresses and skirts, all rated as excellent. The reviews indicate satisfaction with the style, comfort, and quality of these items, suggesting that the customer values fashionable yet practical clothing. Our recommended item, the Aamilife women's 3/4 sleeve cropped cardigan, aligns perfectly with this trend, offering another versatile piece that can complement dresses and skirts. Additionally, the cardigan's open front design and casual nature suggest it would pair well with the other items already in the customer's wardrobe. Given the consistent theme of purchasing stylish and comfortable women's apparel, it is highly likely that the customer sees the cardigan as a valuable addition to their collection. The previous purchases demonstrate a preference for items that enhance wardrobe options, making the cardigan a logical choice. Overall, the combination of style, versatility, and alignment with previous purchases strongly supports the likelihood of this being the next item purchased. </think>
[Recommended Item]
<item_nm> Aamilife women's 3/4 sleeve cropped cardigans sweaters jackets open front short shrugs for dresses </item_nm>

**Rec-SAVER**

[Rationale]
<think> The customer has shown a clear preference for women's clothing items, particularly those that are comfortable and versatile, as evidenced by their purchases of maxi dresses and skirts, all rated as excellent. The reviews indicate satisfaction with the quality and fit of these items, suggesting that the customer values practicality and style in their wardrobe choices. Our recommended item, the Levi's men's 501 original fit jeans, aligns with this trend of seeking reliable and comfortable clothing, even though it is a men's item. The customer may appreciate the classic style and versatility of Levi's jeans, similar to how they found value in their previous purchases. Additionally, the jeans are described as having a relaxed fit, which could appeal to the customer's desire for comfort. Thus, despite being a different category, the choice reflects a broader interest in quality apparel that suits their lifestyle. Overall, the logical connection lies in the customer's established preference for well-made, comfortable clothing. </think>
[Recommended Item]
<item_nm> Levi's men's 501 original fit jeans </item_nm>

**SDPO**

[Rationale]
<think> The customer has shown a clear preference for women's clothing that is stylish, comfortable, and often features unique designs, as evidenced by their purchases of various maxi dresses and skirts, all rated as excellent. The reviews indicate satisfaction with the quality and aesthetic appeal of these items, suggesting that the customer values fashionable yet functional pieces. The recommended item, a humorous graphic t-shirt, aligns with this trend of seeking unique and expressive clothing. Additionally, the customer has previously purchased items that reflect personality, such as the colorful scarves and dresses. The funny nature of the shirt may resonate with the customer's sense of humor, making it appealing as a casual wear option. Overall, the combination of style, comfort, and personal expression makes the t-shirt a fitting choice for the customer. </think>
[Recommended Item]