# OpenReview forum: "More Than What Was Chosen: LLM-based Explainable Recommendation Beyond Noisy User Preferences"
_ICLR.cc/2026/Conference — ICLR 2026 Poster_

### Official Review · Reviewer_AZEL · 2025-10-15

**Soundness:** 3
**Presentation:** 2
**Contribution:** 2
**Rating:** 4
**Confidence:** 4

**Summary:**

The author introduces C-APO, a large-language-model-based recommender framework that unifies Revealed Preference (RP)—the user’s actual selections—with a newly defined Coherent Preference (CP) that measures logical and causal consistency with past interactions.

**Strengths:**

The paper presents a very interesting and well-executed idea. I like that it goes beyond the traditional Revealed Preference view and tries to model Coherent Preference, which captures how consistent a user’s choice is with their history. That’s a fresh perspective, and it makes intuitive sense for real-world noisy data.

The paper doesn’t just show better HR or NDCG—it also includes online A/B testing and a rationale-quality evaluation, which makes the contribution more convincing.

**Weaknesses:**

My main concern lies in how stable and reliable the LLM-generated coherence scores are. Since large language models are inherently stochastic, the same prompt could give slightly different scores on different runs or across model versions. The paper doesn’t discuss how this variability might affect training consistency or bias.

Relatedly, because the same or similar LLMs are used both to generate the rationales and to train the model, there’s a risk of evaluation bias or circular reasoning. It’s not clear how independent the “LLM-as-a-Judge” step is from the model being optimized. Even though human validation is mentioned, the stability and generalizability of those coherence judgments still need stronger evidence.

The smaller concern is scalability. The process of generating and scoring rationales for triplets is expensive, and it’s not obvious how feasible it is to apply at industrial scale. Finally, while the authors describe Coherent Preference as “causal,” it’s more of a semantic or logical coherence measure rather than explicit causal reasoning. Clarifying this point would make the conceptual framing more accurate.

**Questions:**

Since coherence scores are generated by an LLM, how do the authors ensure consistency across runs? LLM outputs can vary depending on temperature, prompt phrasing, or even random seeds. Were the scores averaged over multiple samples, or was the generation deterministic? Some empirical evidence of score stability would strengthen the paper.

The SBERT-based calibration step (Eq. 7) is interesting, but it’s not clear how much it actually changes or stabilizes the raw LLM scores.

The triplet-based data construction is computationally heavy. How practical is it to extend this method to millions of users or large catalogs? Is there any plan for lightweight CP scoring or distillation to make it more scalable?

---

> ### Author Response · Authors · 2025-11-18
> **Response to Reviewer AZEL**
>
> ###### We sincerely appreciate the reviewers’ thoughtful comments and insightful questions. We hope our responses satisfactorily address the concerns raised.
>
> ### **AZEL-W1**
> > *Stability of LLM coherence scores.*
>
> ###### For scoring tasks, we used a deterministic setup with temperature=0, following common practice in LLM-as-a-judge literature. We intentionally adopt this conservative setting because coherence scores act as supervisory signals; introducing stochasticity would inject noise directly into the optimization objective and undermine training stability. We agree with the reviewer that this design choice should have been stated more explicitly, and we have clarified this point in the revised manuscript (line 811, **Appendix §D. Triplet Rationale Dataset Construction Recipe**).
>
>
> ---
> ### **AZEL-W2**
> > *LLM-as-a-Judge circularity concerns.*
>
> ###### We clarify that the ***three LLMs*** used in our pipeline are intentionally separated:
>
> * ###### ***Rationale generation***: gemma-3-it-27B
>
> * ###### ***Training backbone***: gemma-3-it-4B
>
> * ###### ***LLM-as-a-Judge for Rationale Quality***: gpt-4o
>
> ###### Because these models come from different sizes and even different model families, the coherence scoring model is not evaluating itself, and the judge is fully independent from the model being optimized.
>
> ###### To further validate independence, we added results in Section 4.3 showing that replacing the backbone with ***Qwen-3-4B-Instruct*** yields nearly identical performance. This demonstrates that C-APO does not rely on any backbone-specific alignment and that the coherence-based supervision generalizes across architectures. We have now made this independence explicit in the revised manuscript (line 458, **§4.3. Performance Evaluation**).
>
>
>
>
>
>
> ---
> ### **AZEL-W3**
> > *Scalability of triplet construction (Same concern as XFSC-W2).*
>
>
> ###### &nbsp;&nbsp; 1. Operational Cost Perspective.
>
> * ###### In recommender systems where rationales must be shown to users, manually authoring and maintaining rationale templates for each persona or segment can incur significant human-resource overhead—often exceeding the cost of LLM-based dataset construction. C-APO automates this process and therefore reduces long-term operational burden.
>
> ###### &nbsp;&nbsp; 2. Practical Feasibility at Industrial Scale.
>
> * ###### Despite the computational cost, C-APO has already been deployed at industrial scale. Our service serves over 30 million users, yet in practice, a curated subset of only ~100K users (≈300K triplets in C-APO) was sufficient to outperform production ML models trained on million-user datasets. This demonstrates that C-APO benefits more from data quality than data volume.
>
> ###### &nbsp;&nbsp; 3. Scalability Through Lightweight Distillation LLM.
>
> * ###### To further improve scalability, we are developing a lightweight distillation LLM (~4B) that approximates a SOTA LLM (e.g., GPT-4o) at much lower cost, enabling more frequent dataset reconstructions. This direction is now noted in **§6. Conclusion and Future Work**.
>
>
>
> ---
> ### **AZEL-W4**
> > *Clarifying the meaning of "causal" in CP.*
>
> ###### We agree with the reviewer’s point. In the revised manuscript, we have ***removed*** all uses of the term ***causal*** when describing Coherent Preference. To avoid any overclaiming, we now refer to CP strictly as a form of semantic or logical coherence, which more accurately reflects its role in our method.
>
>
>
>
>
> ---
> ### **AZEL-Q1**
> > *Effect of SBERT-Based Calibration.*
>
> ###### The effect of the SBERT-based calibration is already illustrated in Figure 7. As shown in the figure, the calibrated weights display systematic deviations from the raw LLM coherence scores within each $\Delta s$ bin, confirming that the auxiliary text encoder meaningfully adjusts and stabilizes it.
>
> ###### Please also refer to “EkyG-W1: Ablation to isolate the effect of SBERT calibration” for a detailed analysis of the contribution of the SBERT-based calibration module.
>
>
> ---
> ### **AZEL-Q2**
> > *Scalability and Lightweight Scoring.*
>
> ###### Regarding lightweight coherence score generation, we agree that this is an important direction. In fact, we are currently developing a distilled generation model that approximates the coherence score and rationale generating LLM as part of our ongoing effort toward a unified next-generation recommender system. This direction aims to substantially reduce scoring cost while preserving the coherence-judgment capability needed by C-APO. We have also added this point explicitly to the revised manuscript as part of **§6. Conclusion and Future Work**.

---

> > ### Author Response · Authors · 2025-11-26
> > **Official Comment by Authors to Reviewer AZEL**
> >
> > ##### Dear reviewer, as the end of the discussion period approaches, we wanted to reach out and check whether you have any additional questions or points you would like us to clarify. We are happy to address any further follow-up, particularly regarding the broader implications of our findings. Thank you again for taking the time to consider our rebuttal and the substantial revisions we made to the manuscript. Your comments were highly valuable and played an important role in shaping this revision.

---

### Official Review · Reviewer_VjMo · 2025-10-26

**Soundness:** 3
**Presentation:** 3
**Contribution:** 3
**Rating:** 6
**Confidence:** 3

**Summary:**

The paper introduces the concept of coherent preference to complement the traditional principle of revealed preference. The paper proposed a LLM-Rec framework that jointly optimizes RP and CP by constructing an ordered preference data for optimization.

**Strengths:**

1. The approach of inferring intrinsic preferences by reasoning over users’ choices with an LLM is well-motivated, and the mathematical derivation of the optimization is reasonably solid.

2. The authors further propose dynamically adjusting the coherence score to accommodate scenarios with high data noise, where unselected items might actually reflect stronger user interest if available—an aspect often overlooked in existing methods.

3. Extensive and convincing experiments.

**Weaknesses:**

1. The deployment of the LLM appears to substantially increase computational cost, especially when the user interaction history is long. It is mentioned that the LLM is prompted to output a rationale for the user’s choice and a coherence score. This suggests that the overall computational cost could be quite high.

2. Regarding the use of an LLM to generate the coherence score, how much previous interaction history is required? The paper should elaborate on how the length of this history affects the quality of the generated rationales and whether having too few interactions could negatively impact performance. This information would be useful for designing a filtering method to select high-quality training data.

3. A readily foreseeable limitation of the work is the potential “echo chamber” effect, where noisy but exploratory user actions may be smoothed out by the model. The authors might consider discussing this issue further and suggesting possible ways to mitigate it.

**Questions:**

Please see the previous section.

---

> ### Author Response · Authors · 2025-11-18
> **Response to Reviewer VjMo**
>
> ###### We appreciate the considerate reviews and raising important questions. We hope our responses address the reviewer's concerns.
>
> ### **VjMo-W1**
> > *Computational cost of LLM rationale generation.*
>
> ###### We agree that LLM-based recommenders can be more computationally expensive than traditional CF models in the production stage. However, several design choices in C-APO were purposefully made to ensure practical deployability.
>
> ###### &nbsp;&nbsp; 1. Our backbone is a 4B-parameter LLM specifically chosen for production constraints.
>
> * ###### As documented in the manuscript, we adopted a 4B-parameter LLM because larger models would not satisfy service-level latency requirements. This 4B model allowed us to keep inference cost within production budgets while still offering strong reasoning capability.
>
> * ###### **Table 3** shows that our deployed system achieves **138ms per recommendation**, including rationale generation (with four H100 GPUs). This meets the latency constraints of our commercial service, confirming that the computational footprint is in fact compatible with large-scale deployment.
>
> ###### &nbsp;&nbsp; 2. Business impact outweighs the additional cost.
>
> * ###### Although generating rationales introduces some marginal overhead, our online A/B test demonstrates a 1.60x increase in CTR and, more importantly, a 1.29× increase in purchase conversion rate (CVR) over the ML baseline (**§4.6 Online A/B Test**). These gains translate directly into meaningful revenue improvements, which more than justify the additional computation.
>
> * ###### Moreover, in recommendation settings where rationales are not required, the system can simply disable rationale generation through instruction-level control during decoding, thereby eliminating most of the associated computational cost.
>
>
>
>
>
>
>
>
>
> ---
> ### **VjMo-W2**
> > *Effect of history length on rationale quality.*
>
> ###### This analysis has been added to the revised manuscript (**§Appendix D. Triplet Rationale Dataset Construction Recipe**).
>
> ###### &nbsp;&nbsp; 1. Training-time filtering in the original submission.
>
> * ###### As specified in our original submission, users with ***fewer than three interactions*** were already excluded from the training dataset to avoid extremely sparse histories.
>
> ###### &nbsp;&nbsp; 2. Additional analysis on extremely short histories (1–2 interactions).
>
> * ###### To more directly address the reviewer’s concern, we additionally conducted an experiment in which we generated ***rationales*** for users with only 1–2 interactions (sampled from approximately 2,000 instances in the raw Amazon review data) and evaluated their quality using the same 0–3 protocol described in Section 4.4. As expected, histories of length 1–2 produced substantially lower-quality rationales, as shown in the table below.
>
> * ###### These findings confirm that extremely short histories indeed lead to unreliable coherence judgments. Based on this, we now explicitly recommend using history length as a filtering criterion when constructing high-quality training data.
>
> | **history length** | **count** | **mean**   | **std**    |
> |:---------------:|:---------:|:----------:|:----------:|
> |      1-2       |   998     | ***2.20***   | 0.81  |
> |      3–7       |   538     | 2.68   | 0.82   |
> |      8–11      |   142     | 2.70   | 0.81   |
> |       12+       |   220     | 2.76   | 0.71   |

---

> > ### Author Response · Authors · 2025-11-18
> > **Response to Reviewer VjMo - 2**
> >
> > ### **VjMo-W3**
> > > *Potential echo-chamber effects.*
> >
> >
> > ###### We understand the reviewer’s concern as follows: because C-APO is likely to increase the likelihood of history-consistent items, the model might over-anchor on past behaviors, potentially smoothing out exploratory actions and reducing recommendation diversity (i.e., an echo-chamber or low-serendipity effect). We added the related analysis in the **Appendix §I. Serendipity Analysis**.
> >
> > ###### &nbsp;&nbsp; 1. Effect of the hyperparameter $\beta$.
> >
> > * ###### As discussed in **§4.5 Ablation Study**, the hyperparameter $\beta$ controls the strength of policy updates during C-APO alignment. Large values of $\beta$ tend to suppress exploratory behavior, whereas moderate values preserve recommendation diversity. Empirically (Figure 6), settings with $\beta \le 1$ achieve the best balance between accuracy and serendipity.
> >
> > ###### &nbsp;&nbsp; 2. Empirical verification via Serendipity analysis.
> >
> > * ###### To directly evaluate whether C-APO reduces exploratory behavior, we compared serendipity scores between our model and the SFT-only model (i.e., before applying C-APO alignment). We use the following standard formulation:
> >
> >   * ######  $\text{Serendipity}(u)= \frac{| R_u \cap T_u \setminus P |}{|R_u|},$ where $R_u$ denotes recommendations, $T_u$ true interactions, and $P$ globally popular items (@50). This metric measures how often a model surfaces relevant yet non-popular items—higher scores indicate stronger novelty-oriented behavior.
> >
> > * ###### C-APO achieves a serendipity score of 0.30, compared to 0.25 for the SFT-only model. This indicates that C-APO recommends more diverse and less popularity-biased items while still maintaining relevance, providing empirical evidence that the method mitigates echo-chamber tendencies rather than reinforcing them.

---

> > > ### Author Response · Authors · 2025-11-26
> > > **Official Comment by Authors to Reviewer VjMo**
> > >
> > > ##### Dear reviewer, thank you again for taking the time to reassess our work. As the discussion phase enters its final days, we wanted to check whether you have any additional questions or points that you would like us to clarify. We hope that the revisions—along with the substantial additions to the experimental results in both the main paper and the appendix—address your concerns and improve the clarity and contribution of the manuscript.

---

### Official Review · Reviewer_EkyG · 2025-10-30

**Soundness:** 3
**Presentation:** 3
**Contribution:** 2
**Rating:** 4
**Confidence:** 4

**Summary:**

The paper argues that RP (using observed choices) is often noisy and proposes CP: ranking items by their logical/causal consistency with a user’s history (measured via LLM-generated rationales and 7 “coherence” scores). It introduces C-APO, which extends a DPO objective to a triplet ordering (chosen, hard-rejected, asy-rejected) and reweights updates based on RP-CP agreement/conflict using a SBERT-based calibration module. Experiments on multiple Amazon domains show gains over a varietyo f baselines and an online A/B test reports significant CTR gains over a production ML model.

**Strengths:**

- Clear motivation adn evidence of RP -CP conflict
- Principled objective unifying triplet ordering with adaptive weightswith transparent gradient view.
- Thorough empirical sweep across five domains with many baselines, reproducibility steps and code/data link.
- Online A/B provides an initial real-world signal with latency considerations

**Weaknesses:**

- As acknoledged by the authors, human agreement with LLM coherence is positive but imperfect (ρ=0.71), and the paper itself treats LLM scores as noisy. CP also sometimes ranks a rejected item above the chosen one. This is significant as CP scores directly control learning pressure via weights. Residual mis-scoring can redirect gradients and produce brittle behavior in edge cases (e.g., sparse histories). This does not invalidate results but lowers confidence that gains generalize when the coherence signal weakens. More per-domain noise analyses might mitigate this concern.

- The A/B study shows a large CTR lift with impressions/clicks and a z-test, yet lacks confidence intervals and downstream KPIs. The deployment is a pilot over a 2-month window. For claims of practical utility, reviewers need assurance that lifts are not artifacts of short-term novelty, catalog skew, or traffic allocation quirks, especially since the method changes not only ranking but rationales surfaced to users. Additional transparency would substantiate impact claims. Can you please report CTR CIs, per-user variance, and any downstream metrics (e.g., dwell, purchase, satisfaction proxy). Also clarify any holdout/novelty controls.

- All offline datasets are Amazon text domains. The approach conceptually targets broad recommenders (video, image, music), but no multimodal or cold-start setting is shown. The coherence construct likely depends on modality. Showing at least one non-text domain would strengthen the generality claim.

- The paper shows Base < SFT < DPO/PL < C-APO but does not include “C-APO (no SBERT calibration)” or “PL + calibration (no conflict-aware weighting)”. Without isolating the calibration block (Eq. 7), reviewers cannot attribute improvements to the central claimed idea (conflict-aware weighting) versus the auxiliary scoring stabilizer. This affects interpretability of the contribution and the portability of the method to settings where the calibration model might behave differently. Can you add an ablation for C-APO without the SBERT-gate and/or PL + calibration but no conflict-aware weighting? This would clarify which piece drives the gains.

**Questions:**

- Please see the weaknesses section.
- Were the annotation guidelines and instruction provided for the annotators disclosed? How many annotators were involved? Were they diverse in terms of demographics? What was their background in annotation? Additionally, what measures were taken to validate the quality and consistency of the annotations, and ensure the reliability of the labeled data?

---

> ### Author Response · Authors · 2025-11-18
> **Response to Reviewer EkyG**
>
> ###### We appreciate the constructive feedback. We hope our responses address the reviewer's concerns.
>
> ### **EkyG-W1**
> > *Ablation to isolate the effect of SBERT calibration.*
>
> ###### &nbsp;&nbsp; 1. C-APO w/o SBERT calibration (new variant).
>
> * ###### We added a new ablation, and its results are included in Figure 5 and discussed in revised **§4.5. Ablation Study** of the revised manuscript. This variant removes the SBERT-based calibration module and uses raw LLM coherence scores directly (pairwise difference → sigmoid). This allows us to test how much of the performance gain comes purely from using uncalibrated LLM scores, without relying on an additional text encoder (SBERT) to stabilize or adjust them. As shown in the table below, this variant performs better than PL but worse than the full C-APO, showing that:
>
>   * ###### using LLM coherence signals alone already helps, but
>   * ###### SBERT-based calibration provides the main performance boost by correcting noise in the LLM scores.
>
>
> | **Domain**   | **Model** | **HR@1** | **NDCG@5** |
> |--------------|-----------|----------|------------|
> | **Fashion**  | PL        | 5.31     | 14.32      |
> |              | Ours w/o SBERT   | 6.05     | 16.99      |
> |              | Ours      | **9.47** | **18.87**  |
> | **Grocery**  | PL        | 6.20     | 12.45      |
> |              | Ours w/o SBERT   | 6.11     | 13.36      |
> |              | Ours      | **6.48** | **13.21**  |
> | **Scientific** | PL      | 4.92     | 12.38      |
> |              | Ours w/o SBERT   | 7.94     | 15.14      |
> |              | Ours      | **12.22**| **17.97**  |
> | **Clothing** | PL        | 5.74     | 14.24      |
> |              | Ours w/o SBERT   | 6.51     | 14.30      |
> |              | Ours      | **7.11** | **15.09**  |
> | **Health**   | PL        | 4.04     | 8.48       |
> |              | Ours w/o SBERT   | 3.15     | 9.68       |
> |              | Ours      | **4.83** | **11.73**  |
>
>
>
>
>
>
>
> ---
> ### **EkyG-W2**
> > *A/B test lacks confidence intervals and downstream KPIs.*
>
> ###### In the revised manuscript (**§4.6. Online A/B Test**, Table 3), we substantially expand our analysis to address all concerns, including novelty controls, confidence intervals, downstream KPIs, and longer-term stability, as shown below Table.
>
> ###### &nbsp;&nbsp; 1. Extended evaluation window and stable traffic allocation.
>
> * ###### Users were assigned to buckets via a stable user-level hash with no overlap, and each user remained in a fixed arm throughout the experiment. Since the deployment continued internally after paper submission, we now report 18 weeks (August–November 2025) of live data (up from the original 7-week window). This longer horizon confirms that the improvements are not short-lived artifacts of novelty, catalog effects, or traffic imbalance.
>
> ###### &nbsp;&nbsp; 2. Explicit novelty control via an SFT-based baseline.
>
> * ###### To isolate the effect of rationale exposure from the effect of our method, we added a fourth bucket (Nov 13~ ;during one week): an SFT-based model that generates recommendations and rationales. Because this model exposes rationales just like ours, it serves as a direct novelty-control baseline. Even under this stricter comparison, C-APO achieves a statistically significant CTR gain (+1.47\%p; 15.03\% vs. 13.56\%; $Z$ = 3.20, $p <$ 0.001; 95\% CI = [0.57 \%p, 2.37 \%p]). This demonstrates that the observed improvements do not stem from the novelty of showing rationales.
>
> ###### &nbsp;&nbsp; 3. Confidence intervals.
>
> * ###### We additionally report 95\% confidence intervals for the CTR differences in the two key comparisons—Ours vs. ML and Ours vs. SFT-based model in the revised manuscript. Our model achieved a +60.88\% CTR lift over the ML baseline (one-sided two-proportion z-test: $Z = 39.42, p < 0.001$; 95\% CI for the absolute CTR difference: [5.40\%, 5.97\%]). It also showed a +1.47\%p improvement over the SFT-based model (15.03\% $vs$. 13.56\%; $Z = 3.20, p < 0.001$; 95\% CI: [0.57\%p, 2.37\%p]).
>
> ###### &nbsp;&nbsp; 4. Additional downstream KPIs.
>
> * ###### We now include conversion rate (CVR)—the primary downstream business KPI, defined as the number of purchases per impression and directly tied to revenue—as part of the online evaluation. Reporting both CTR and CVR over an extended 18-week period provides stronger evidence that the gains are robust, stable, and practically meaningful.
>
> | **Model**  | **# Impression** | **# Clicks** | **CTR**  | **CVR**  | **Latency**    |
> |-----------|------------------|--------------|----------|----------|----------------|
> | Random    | 180.08K          | 17.90K       | 9.90%    | 1.58%    | 94ms/call      |
> | ML model  | 127.83K          | 11.94K       | 9.34%    | 2.01%    | 120ms/call     |
> | SFT (***New***)       | 6.50K            | 0.88K        | 13.56%   | 1.81%    | 132ms/call     |
> | **Ours**  | **79.50K**       | **11.95K**   | **15.03%** | **2.60%** | **138ms/call** |

---

> ### Author Response · Authors · 2025-11-18
> **Response to Reviewer EkyG - 2**
>
> ### **EkyG-W3**
> > *Lack of Cold-Start Analysis and Multimodal Generalization*
>
> ###### &nbsp;&nbsp; 1. Cold-start and zero-shot domain transfer.
>
> * ###### To address the reviewer’s question regarding cold-start scenarios, we added new zero-shot experiments where the model is trained on one domain and evaluated on a related but unseen domain (**Appendix §H.1. Cold-Start and Zero-Shot Domain Transfer**).
>
> * ###### Specifically, we trained C-APO on Amazon Fashion dataset and evaluated on Amazon Beauty, and trained on Musical Instruments dataset and evaluated on CDs \& Vinyl. In the Fashion→Beauty transfer, C-APO even outperformed a CF-Rec model trained directly on Beauty across all metrics (HR@1, HR@5, NDCG@5). In the Musical Instruments→CDs \& Vinyl transfer, C-APO also achieved competitive performance. These results indicate that C-APO learns a domain-transferable representation of user preference consistency, enabling effective cold-start recommendation.
>
>
> | **Domain** | **Amazon Beauty** |  |  | **CDs and Vinyl** |  |  |
> |-----------|-------------------|----|----|-------------------|----|----|
> | **Models** | HR@1 | HR@5 | N@5 | HR@1 | HR@5 | N@5 |
> | SASRec | 2.90 | 16.12 | 9.18 | 3.26 | 16.34 | 9.61 |
> | BERT4Rec | 3.26 | 15.04 | 9.01 | 3.14 | 14.96 | 8.89 |
> | GRU4Rec | 3.26 | 17.57 | 10.51 | 4.29 | 24.00 | 13.73 |
> | FOSSIL | 1.81 | 13.95 | 7.84 | 3.72 | 15.74 | 9.59 |
> | NextItNet | 2.72 | 14.67 | 8.45 | 3.68 | 15.65 | 9.57 |
> | TransRec | 3.62 | 17.21 | 10.33 | 2.93 | 14.90 | 8.69 |
> | SINE | 3.80 | 15.04 | 9.26 | 4.64 | 18.04 | 11.23 |
> | Rec-SAVER (Unseen) | 8.56 | 31.51 | 19.69 | **10.16** | 27.84  | 19.40  |
> | **Ours (Unseen)** | **19.49** | **40.62** | **30.81** | 10.08 | **28.44** | **19.66** |
>
>
> ###### &nbsp;&nbsp; 2. Experiments on Non-Amazon Domains (MovieLens).
>
> * ###### We agree that C-APO can naturally extend to broader recommendation scenarios, including multimodal settings. While text signals are widely available across most platforms, many datasets lack consistent image or audio features, and multimodal models often incur substantially higher generation, training, and inference costs—making them less practical for large-scale commercial deployment. Nevertheless, multimodal extensions hold potential for improving recommendation performance, and we plan to explore this direction in future work (***§6. Conclusion and Future Work***).
>
> * ###### To demonstrate that C-APO is not limited to free-form review text, we also evaluated the model on the MovieLens dataset (**Appendix §H.2. Experiments on Non-Amazon Domains**), where items contain only short user-generated tags rather than rich textual descriptions. Despite this structural difference, our model consistently outperformed most baselines (see the table below), mirroring the trends observed in the five Amazon domains. This confirms that the core principles of our approach generalize beyond Amazon-style text datasets.
>
> | **Model (MovieLens)**  | **BERT4Rec** | **GRU4Rec** | **NextItNet** | **SINE** | **STAMP** | **Ours** |
> |-----------:|-------------:|------------:|--------------:|---------:|---------:|---------:|
> | **HR@1**   | 2.26 | 2.51 | 2.25 | 1.93 | 2.08 | **2.81** |
> | **HR@5**   | 12.84 | 13.43 | 10.41 | 11.60 | 9.86 | **13.13** |
> | **NDCG@5** | 7.38 | 7.86 | 6.24 | 6.61 | 5.88 | **8.10** |
>
>
>
>
>
>
>
> ---
> ### **EkyG-W4**
> > *Noise in coherence score.*
>
>
> ###### &nbsp;&nbsp; 1. Noise in LLM-generated coherence scores is handled explicitly.
>
> * ###### While LLM-generated coherence scores may contain some inherent noise, C-APO explicitly mitigates this through ***confidence-weighted adaptive weight*** rather than treating coherence scores as hard labels, directly addressing the reviewer’s concern about gradient misdirection.
>
> ###### &nbsp;&nbsp; 2. Robustness Under Sparse Histories.
>
> * ###### To further verify robustness, we evaluated C-APO on users with sparse histories (1–3 interactions)—the setting most vulnerable to noisy coherence judgments. Even in this challenging regime, C-APO consistently outperformed the SFT-only model (i.e., before applying C-APO), and in some cases matched or exceeded the performance observed in longer-history bins.
>
> * ###### These results indicate that our calibration module effectively stabilizes noisy LLM-generated scores, allowing the model to remain reliable even when user histories are extremely short, thereby alleviating the gradient-misdirection issue raised by the reviewer.
>
>
> | **History Length** | **SFT HR@1** | **SFT HR@5** | **SFT N@5** | **Ours HR@1** | **Ours HR@5** | **Ours N@5** |
> |----------------:|------------:|------------:|------------:|-------------:|-------------:|-------------:|
> | **1–3**         | 3.76        | 17.29       | 10.75       | **6.42**     | **20.78**    | **13.99**    |
> | **4–7**         | 3.66        | 16.82       | 10.30       | **6.35**     | **19.87**    | **13.51**    |
> | **8–11**        | 3.29        | 18.22       | 10.80       | **5.12**     | **18.34**    | **11.97**    |

---

> > ### Author Response · Authors · 2025-11-18
> > **Response to Reviewer EkyG - 3**
> >
> > ### **EkyG-Q1**
> > > *Annotation protocol (guidelines, annotator background, diversity).*
> >
> > ###### In the revised manuscript, we provide a clearer and more complete description of the annotators, guidelines, and quality-control procedures (added in **Appendix §D. Triplet Rationale Dataset Construction Recipe**).
> >
> > ###### &nbsp;&nbsp; 1. Annotators.
> >
> > * ###### A total of 7 domain experts participated in the evaluation: 1 service marketer (female, age 34), 2 data engineers (male, ages 41 and 29), and 4 data scientists (male, ages 37, 33, 31, and 30). This group reflects diverse professional roles across product, engineering, and modeling, all directly involved in the recommender system context.
> >
> > ###### &nbsp;&nbsp; 2. Annotation guidelines.
> >
> > * ###### We have added the full set of instructions provided to human annotators in in **Appendix §D**. Annotators were shown the same prompt used for LLM-based coherence score generation.
> >
> > * ###### In addition, annotators received a separate human-oriented instruction sheet explaining how to judge the coherence between the user’s history and the recommended item, along with several few-shot examples to calibrate expectations.
> >
> > ###### &nbsp;&nbsp; 3. Quality and consistency control.
> >
> > * ###### All annotators scored a shared subset of items, enabling measurement of inter-annotator agreement. For cases where annotators’ scores differed by more than 4 points on the 7-point scale, a consensus procedure was conducted to resolve discrepancies. These measures ensure the reliability and consistency of the collected labels.

---

> > > ### Author Response · Authors · 2025-11-26
> > > **Official Comment by Authors to Reviewer EkyG**
> > >
> > > ##### Dear reviewer, as the end of the discussion period approaches, we wanted to reach out and check whether you have any additional questions or points you would like us to clarify. Thank you again for taking the time to consider our rebuttal and the substantial changes we made to the manuscript. Your comments were highly valuable and directly informed the revisions. We hope that the improvements—along with the expanded experimental results in both the main paper and the appendix—address your concerns and strengthen the contribution.

---

> > > > ### Comment · Reviewer_EkyG · 2025-11-27
> > > > **Response**
> > > >
> > > > I thank the authors for the detailed response and revision. My main concerns have been largely addressed, and I have therefore upgraded my score.

---

> > > > > ### Author Response · Authors · 2025-11-28
> > > > > **Official Comment by Authors**
> > > > >
> > > > > ##### Thank you very much for taking the time to reassess our work and for the positive update to your score. We truly appreciate your constructive feedback throughout the discussion phase—it significantly improved the clarity and quality of the paper. If there are any remaining points you feel we should refine further in the revised version, we would be very happy to incorporate them.

---

### Official Review · Reviewer_XFSC · 2025-10-31

**Soundness:** 3
**Presentation:** 2
**Contribution:** 3
**Rating:** 6
**Confidence:** 4

**Summary:**

Traditional recommender systems, including those using LLMs, rely on Revealed Preference (RP), assuming user actions reflect true interests. However, user behavior is often noisy and inconsistent, leading to flawed recommendations and unpersuasive rationales.

This paper introduces Coherent Preference (CP), a concept prioritizing items that are logically and causally consistent with a user's entire interaction history, not just isolated choices.

Building on this, the authors propose Conflict-Aware Direct Preference Optimization (C-APO), an LLM-based framework that jointly optimizes for both RP and CP. C-APO uses a unified ranking system that combines signals from both preferences. Critically, it adaptively reconciles agreements and conflicts between RP and CP, allowing it to better capture genuine user intent.

On the Amazon Review dataset, C-APO outperformed approximately 20 state-of-the-art baselines in both recommendation performance and rationale quality. A real-world deployment confirmed its practical effectiveness.

**Strengths:**

The idea proposed in this paper is easy to understand, direct, and effective. However, the cost of dataset construction might need to be considered.

The paper provides some theoretical analysis and proofs.

The experiments are very thorough, for example, comparing against as many as 20 baselines and using datasets from multiple domains. The experimental results also demonstrate the effectiveness of the proposed model.

The code and dataset are publicly available.

**Weaknesses:**

Figure 2 and the paragraph that references it (i.e., the paragraph starting around line 048) do not seem consistent. I could not fully understand what the authors were trying to illustrate with this paragraph and Figure 2.

The paper might be somewhat difficult for readers who are not very familiar with related work. For instance, some terms like "rejected item" lack an intuitive explanation when they first appear.

**Questions:**

Regarding line 242, how is the conflict-aware reward weight w_{i,j} defined? Is it a function with g_i - g_j as input? Is it learned, or is it predefined as a constant?

---

> ### Author Response · Authors · 2025-11-18
> **Response to Reviewer XFSC**
>
> ###### We appreciate the insightful comments and positive support with constructive feedback. We hope our responses address the reviewer's concerns.
>
> ### **XFSC-W1**
> > *Inconsistency between Figure 2 and the referencing paragraph.*
>
> ###### The example is intended to illustrate an RP–CP conflict—a user with a history in fantasy and romance suddenly choosing an action movie, which may reflect shared-account usage or temporary promotional effects rather than a true preference shift. We have **revised both the text (line 48) and Figure 2 in §1. Introduction** to ensure full consistency.
>
> ###### &nbsp;&nbsp; 1. Correction of genre description.
>
>   * ###### The paragraph beginning at line 48 described the user’s history as ***melodrama***, whereas Figure 2 actually depicts ***fantasy*** and ***romance*** genre movies. We have updated the manuscript text so that the genres now match the figure.
>
> ###### &nbsp;&nbsp; 2. Correction of caption of Figure 2.
>
>   * ###### We revised the caption of Figure 2 to use generic genre labels (Action or Romance) instead of specific movie titles such as Kingdom or About Time.
>
>
>
> ---
> ### **XFSC-W2**
> > *Issues with dataset construction cost.*
>
> ###### &nbsp;&nbsp; 1. Operational Cost Perspective.
>
> * ###### In recommender systems where rationales must be shown to users, manually authoring and maintaining rationale templates for each persona or segment can incur significant human-resource overhead—often exceeding the cost of LLM-based dataset construction. C-APO automates this process and therefore reduces long-term operational burden.
>
> ###### &nbsp;&nbsp; 2. Practical Feasibility at Industrial Scale.
>
> * ###### Despite the computational cost, C-APO has already been deployed at industrial scale. Our service serves over 30 million users, yet in practice, a curated subset of only ~100K users (≈300K triplets in C-APO) was sufficient to outperform production ML models trained on million-user datasets. This demonstrates that C-APO benefits more from data quality than data volume.
>
> ###### &nbsp;&nbsp; 3. Scalability Through Lightweight Distillation LLM.
>
> * ###### To further improve scalability, we are developing a lightweight distillation LLM (~4B) that approximates a SOTA LLM (e.g., GPT-4o) at much lower cost, enabling more frequent dataset reconstructions. This direction is now noted in **§6. Conclusion and Future Work**.
>
>
>
>
>
> ---
> ### **XFSC-W3**
> > *Some terms such as “rejected item” lack intuitive explanation.*
>
> * ###### To improve readability, we now introduce key terminology—such as rejected item—earlier in the manuscript with explicit definitions. In particular, we revised the introductory description (line 76 in **§1. Introduction**) of triplet construction to read:
>
>   * ###### ​“For each user, we construct a triplet consisting of the ground-truth chosen item and two unobserved alternatives—items the user did not interact with, which we hereafter refer to as rejected items.”
>
> * ###### For additional clarity, we also note that in DPO-style preference learning, the chosen item corresponds to the positive sample, while rejected items serve as negative samples.
>
>
> ---
> ### **XFSC-Q1**
> > *Clarification on the definition of the reward weight $w_{i,j}$.*
>
>
> ###### Importantly, $w_{i,j}$ is ***not*** a hand-crafted constant nor a simple function of $g_i - g_j$. Rather, it is a ***trainable*** probabilistic weight computed via the calibration module (Eq. 7), consistent with a Thurstone--Mosteller model. In the final formulation, this trainable weight $w_{i,j}$ is ***multiplied by*** the score difference ($g_i - g_j$), ensuring that the calibrated pairwise preference directly scales the effective comparison signal. As suggested, we have updated the manuscript (line 250 in **§3.2. Derivation of C-APO**) to explicitly state near Eq. 7 that the calibration module---including the SBERT-gate and the MLPs producing $\mu$ and $\tilde{\sigma}$---is  ***trainable*** and jointly optimized with the LLM backbone. This clarification has been added to the revised version.
>
>   * ###### ​“We define the conflict-aware reward difference as $w_{i,j}(g_i - g_j)$, where $w_{i,j}$ denotes the ***trainable*** conflict-aware adaptive weight assigned to each item pair $i,j \in \{c,h,e\}$.”

---

> > ### Author Response · Authors · 2025-11-26
> > **Official Comment by Authors to Reviewer XFSC**
> >
> > ##### Dear Reviewer, thank you once again for taking the time to reassess our work. As we approach the end of the discussion phase, we wanted to kindly check whether you have any additional questions or points that you would like us to clarify. We hope that the recent improvements to the manuscript address your concerns and provide a clearer presentation of our contributions.

---

### Author Response · Authors · 2025-11-18
**Summary of the Revised Sections**

***Summary of the Revised Sections***
> Please refer to the relevant sections in the revised manuscript.


**§1. Introduction**
> * (revised) Figure 2 and  line 48: Inconsistency between Figure 2 and the referencing paragraph (XFSC).
> * (revised) line 76: added definition of rejected item (XFSC)

**§3.2. Derivation of C-APO**
> * (revised) line 250: conflict-aware adaptive weight → ***trainable*** conflict-aware adaptive weight (XFSC)

**§4.5. Ablation Study**
> * (new) Added new variants in the ablation study (C-APO w/o SBERT calibration) (EkyG, AZEL)
> * (revised) Figure 5


**§4.6 Online A/B Test**
> * (new) Added the confidence intervals and downstream KPIs. (EkyG)
> * (revised) Table 3


**Appendix §D. Triplet Rationale Dataset Construction Recipe**
> * (new) Added annotation protocol (EkyG)
> * (new) Added the effect of history length on the rationales quality (VjMo)
> * (revised) line 811: deterministic setup for LLM coherence scores (AZEL)

**Appendix §H.1. Cold-Start and Zero-Shot Domain Transfer**
> * (new) Added the cold-start scenario analysis (EkyG)


**Appendix §H.2. Experiments on Non-Amazon Domains**
> * (new) Added experiments on Non-Amazon Domains (MovieLens; EkyG)


**Appendix §I. Serendipity Analysis**
> * (new) Added the serendipity analysis (VjMo)



**§6. Conclusion and Future Work**
> * (new) Lightweight Scoring LLM (XFSC, AZEL)
> * (new) Multi-modality (EkyG)

---

### Meta-Review · Area_Chair_rEFH · 2026-01-09

**Summary:**

The paper introduces Coherent Preference and the C-APO objective to reconcile noisy revealed preferences, with strong offline results against many baselines, public artifacts, and promising online A/B evidence. Main concerns are: reliability/stability of LLM-generated coherence scores (and potential training brittleness), missing ablations to attribute gains (conflict-aware weighting vs. SBERT calibration), scalability/cost and sensitivity to history length, and more complete online-study reporting (CIs, downstream KPIs, controls) plus minor clarity/framing issues (definitions, echo-chamber risk, ...).

Overall, the concerns appear addressable with additional ablations, stability analyses, and clearer reporting provided by the authors.

**Reviewer Concerns:**

I think concerns raised by XFSC, EkyG, and VjMo are addressed.

Regarding AZEL, major concerns about performance stability are addressed by additional results, while minor concerns about scalability are addressed by the authors.

**Reviewer Scores:**

I think the four reviewers might have rated this paper as 6.

---

### Decision · Program_Chairs · 2026-01-26

Accept (Poster)